# QUANTITATIVE APPROXIMATION FOR NEURAL OPERATORS IN NONLINEAR PARABOLIC EQUATIONS

**Takashi Furuya**[1,*]    **Koichi Taniguchi**[2,*]    **Satoshi Okuda**[3]

[1]Doshisha University, `takashi.furuya0101@gmail.com`
[2]Shizuoka University, `taniguchi.koichi@shizuoka.ac.jp`
[3]Rikkyo University, `okudas@rikkyo.ac.jp`
* These authors contributed equally to this work

## ABSTRACT

Neural operators serve as universal approximators for general continuous operators. In this paper, we derive the approximation rate of solution operators for the nonlinear parabolic partial differential equations (PDEs), contributing to the quantitative approximation theorem for solution operators of nonlinear PDEs. Our results show that neural operators can efficiently approximate these solution operators without the exponential growth in model complexity, thus strengthening the theoretical foundation of neural operators. A key insight in our proof is to transfer PDEs into the corresponding integral equations via Duahamel's principle, and to leverage the similarity between neural operators and Picard's iteration—a classical algorithm for solving PDEs. This approach is potentially generalizable beyond parabolic PDEs to a class of PDEs which can be solved by Picard's iteration.

## 1    INTRODUCTION

Neural operators have gained significant attention in deep learning as an extension of traditional neural networks. While conventional neural networks are designed to learn mappings between finite-dimensional spaces, neural operators extend this capability by learning mappings between infinite-dimensional function spaces. A key application of neural operators is in constructing surrogate models of solvers for partial differential equations (PDEs) by learning their solution operators. Traditional PDE solvers often require substantial computational resources and time, especially when addressing problems with high dimensionality, nonlinearity, or complex boundary shape. In contrast, once trained, neural operators serve as surrogate models, providing significantly faster inference compared to traditional numerical solvers. Neural operators are recognized as universal approximators for general continuous operators. However, the theoretical understanding of their approximation capabilities, particularly as solvers for PDEs, is not yet fully developed.

This paper focuses on whether neural operators are suitable for approximating solution operators of PDEs, and which neural operator architectures might be effective for this purpose. The idea of this paper is to define neural operators by aligning them with Picard's iteration, a classical method for solving PDEs. Specifically, by associating each forward pass of the neural operator's layers with a Picard's iteration step, we hypothesize that increasing the number of layers would naturally lead to an approximate solution of the PDE. We constructively prove a quantitative approximation theorem for solution operators of PDEs based on this idea. This theorem shows that, by appropriately selecting the basis functions within the neural operator, it is possible to avoid the exponential growth in model complexity (also called "the curse of parametric complexity") that often arises in general operator approximation, thereby providing a theoretical justification for the effectiveness of neural operators as PDE solvers.

### 1.1    RELATED WORKS

Neural operators were introduced by Kovachki et al. (2023) as one of the operator learning methods, such as DeepONet (Chen & Chen, 1995; Lu et al., 2019) and PCA-Net (Bhattacharya et al., 2021).

Various architectures have been proposed, including Graph Neural Operators (Li et al., 2020b), Fourier Neural Operators (Li et al., 2020a), Wavelet Neural Operators (Tripura & Chakraborty, 2023; Gupta et al., 2021), Spherical Fourier Neural Operators (Bonev et al., 2023), and Laplace Neural Operators (Chen et al., 2024). These architectures have demonstrated empirical success as the surrogate models of simulators across a wide range of PDEs, as benchmarked in Takamoto et al. (2022). In the case of parabolic PDEs, which are the focus of this paper, neural operators have also shown promising results, for example, the Burgers, Darcy, Navier-Stokes equations (Kovachki et al., 2023), the KPP-Fisher equation (Takamoto et al., 2022), the Allen-Cahn equation (Tripura & Chakraborty, 2023; Navaneeth et al., 2024), and the Nagumo equation (Navaneeth et al., 2024).

Universal approximation theorems of operator learning for general operators were established for neural operators (Kovachki et al., 2021; 2023; Lanthaler et al., 2023; Kratsios et al., 2024), Deep-Onet (Lu et al., 2019; Lanthaler et al., 2022), and PCA-net (Lanthaler, 2023). This indicates that these learning methods possess the capabilities to approximate a wide range of operators. However, operator learning for general operators suffers from "the curse of parametric complexity", where the number of learnable parameters exponentially grows as the desired approximation accuracy increases (Lanthaler & Stuart, 2023).

A common approach to mitigating the curse of parametric complexity is to restrict general operators to the solution operators of PDEs. Recently, several quantitative approximation theorems have been established for the solution operators of specific PDEs without experiencing exponential growth in model complexity. For instance, Kovachki et al. (2021); Lanthaler (2023) developed quantitative approximation theorems for the Darcy and Navier-Stokes equations using Fourier neural operators and PCA-Net, respectively. Additionally, Lanthaler & Stuart (2023) addressed the Hamilton-Jacobi equations with Hamilton-Jacobi neural operators. Further studies, such as Chen et al. (2023); Lanthaler et al. (2022); Marcati & Schwab (2023), investigated quantitative approximation theorems using DeepONet for a range of PDEs, including elliptic, parabolic, and hyperbolic equations, while Deng et al. (2022) focused on advection-diffusion equations.

In line with these researches, the present paper also concentrates on restricting the learning operator to the solution operators of specific PDEs, namely parabolic PDEs. However, unlike previous studies, this work leverages the similarity with the Picard's iteration in the framework of relatively general neural operator. This approach is potentially generalizable beyond parabolic PDEs to a range of other equations, including the Navier-Stokes equation, nonlinear dispersive equations, and nonlinear hyperbolic equations, which are solvable by Picard's iteration. See Kovachki et al. (2024) for other directions on the quantitative approximation of operator learning.

## 1.2 OUR RESULTS AND CONTRIBUTIONS

In this paper, we present a quantitative approximation theorem for the solution operator of nonlinear parabolic PDEs using neural operators. Notably, in Theorem 1, we show that for any given accuracy, the depth and the number of neurons of the neural operators do not grow exponentially.

The proof relies on Banach's fixed point theorem. Under appropriate conditions, the solution to nonlinear PDEs can be expressed as the fixed point of a contraction mapping, which can be implemented through Picard's iteration. This contraction mapping corresponds to an integral operator whose kernel is the Green function associated with the linear equation. By expanding the Green function in a certain basis and truncating the expansion, we approximate the contraction mapping using a layer of neural operator. In this framework, the forward propagation through the layers is interpreted as steps in Picard's iteration. Our approach does not heavily rely on the universality of neural networks. More specifically, we utilize their universality only to approximate the nonlinearity, which is represented as a one-dimensional nonlinear function, and hence, the approximation rates obtained in Theorem 1 correspond to the one-dimensional approximation rates of universality. As a result, our approach demonstrates that exponential growth in model complexity of our neural operators can be avoided.

## 1.3 NOTATION

We introduce the notation often used in this paper.

- For $r \in [1, \infty]$, we write the Hölder conjugate of $r$ as $r'$: $1/r + 1/r' = 1$ if $r \in (1, \infty)$; $r' = \infty$ if $r = 1$; $r' = 1$ if $r = \infty$.

- Let $X$ be a set. For an operator $A : X \to X$ and $k \in \mathbb{N}$, we denote by $A^{[k]}$ the $k$ times compositions of $A$ (or the $k$ times products of $A$): $A^{[0]}$ means the identity operator on $X$ and $A^{[k]} := \underbrace{A \circ \cdots \circ A}_{k \text{ times}}$.

- Let $X$ and $Y$ be normed spaces with norm $\| \cdot \|_X$ and $\| \cdot \|_Y$, respectively. For a linear operator $A : X \to Y$, we denote by $\|A\|_{X \to Y}$ the operator norm of $A$: $\|A\|_{X \to Y} := \sup_{\|f\|_X = 1} \|Af\|_Y$.

- Let $X$ be a normed space with norm $\| \cdot \|_X$. We denote by $B_X(R)$ the closed ball in $X$ with center 0 and radius $R > 0$: $B_X(R) := \{f \in X : \|f\|_X \le R\}$.

- For $q \in [1, \infty]$, the Lebesgue space $L^q(D)$ is defined by the set of all measurable functions $f = f(x)$ on $D$ such that $\|f\|_{L^q} := (\int_D |f(x)|^q \, dx)^{\frac{1}{q}} < \infty$ if $q \ne \infty$ and $\|f\|_{L^\infty} := \operatorname{ess\,sup}_{x \in D} |f(x)| < \infty$ if $q = \infty$. For $r, s \in [1, \infty]$, the space $L^r(0, T; L^s(D))$ is defined by the set of all measurable functions $f = f(t, x)$ on $(0, T) \times D$ such that

$$\|f\|_{L^r(0,T;L^s)} := \left\{ \int_0^T \left( \int_D |f(t,x)|^s \, dx \right)^{\frac{r}{s}} dt \right\}^{\frac{1}{r}} < \infty$$

(with the usual modifications for $r = \infty$ or $s = \infty$).

- For $r, s \in [1, \infty]$, the notation $\langle \cdot, \cdot \rangle$ means the dual pair of $L^{s'}(D)$ and $L^s(D)$ or $L^{r'}(0, T; L^{s'}(D))$ and $L^r(0, T; L^s(D))$:

$$\langle u, v \rangle := \int_D u(x)v(x) \, dx \quad \text{or} \quad \langle u, v \rangle := \int_0^T \int_D u(t,x)v(t,x) \, dxdt,$$

respectively.

## 2 LOCAL WELL-POSEDNESS FOR NONLINEAR PARABOLIC EQUATIONS

To begin with, we describe the problem setting of PDEs addressed in this paper. Let $D$ be a bounded domain in $\mathbb{R}^d$ with $d \in \mathbb{N}$. We consider the Cauchy problem for the following nonlinear parabolic PDEs:

$$\begin{cases} \partial_t u + \mathcal{L}u = F(u) & \text{in } (0, T) \times D, \\ u(0) = u_0 & \text{in } D, \end{cases} \tag{P}$$

where $T > 0$, $\partial_t := \partial/\partial t$ denotes the time derivative, $\mathcal{L}$ is a certain operator (e.g. $\mathcal{L} = -\Delta := -\sum_{j=1}^d \partial^2/\partial x_j^2$ is the Laplacian), $F : \mathbb{R} \to \mathbb{R}$ is a nonlinearity, $u : (0, T) \times D \to \mathbb{R}$ is a solution to (P) (unknown function), and $u_0 : D \to \mathbb{R}$ is an initial data (prescribed function). It is important to note that boundary conditions are contained in the operator $\mathcal{L}$. In this sense, the problem (P) can be viewed as an abstract initial boundary value problem on $D$. See Appendix A for examples of $\mathcal{L}$.

As it is a standard practice, we study the problem (P) via the integral formulation

$$u(t) = S_\mathcal{L}(t)u_0 + \int_0^t S_\mathcal{L}(t - \tau)F(u(\tau)) \, d\tau, \tag{P'}$$

where $\{S_\mathcal{L}(t)\}_{t \ge 0}$ is the semigroup generated by $\mathcal{L}$ (i.e. the solution operators of the linear equation $\partial_t u + \mathcal{L}u = 0$). The second term in the right hand side of (P') is commonly referred to as the Duhamel's integral. Under suitable conditions on $\mathcal{L}$ and $F$, the problems (P) and (P') are equivalent if the function $u$ is sufficiently smooth. For example, in the case where $\mathcal{L} = -\Delta$, it follows from the smoothing effect of $S_\mathcal{L}(t)$ that the solution to (P') is a classical solution to (P) (see the argument in the proof of Brezis & Cazenave (1996, Theorem 1)). Even in the case where $\mathcal{L}$ is a more general operator satisfying Assumption 1, a similar argument can be also done by replacing the differentiation in $x$ with the operator $\mathcal{L}$ (together with use of the techniques in the proofs of Iwabuchi et al. (2021, Lemmas 3.5 and 3.10) for instance). The aim of this section is to state the results on local well-posedness (LWP) for (P'). Here, the LWP means the existence of local in time

solution, the uniqueness of the solution, and the continuous dependence on initial data. Its proof is based on the fixed point argument (or also called the contraction mapping argument). These results are fundamental to study our neural operator in Sections 3 and 4. In particular, Propositions 1 and 2 below serve as guidelines for setting function spaces as the domain and range of neural operators in Definition 1 (Section 3) and for determining the norm to measure the error in Theorem 1 (Section 4).

In this paper we impose the following assumptions on $\mathcal{L}$ and $F$.

**Assumption 1.** *For any $1 \leq q_1 \leq q_2 \leq \infty$, there exists a constant $C_{\mathcal{L}} > 0$ such that*

$$\|S_{\mathcal{L}}(t)\|_{L^{q_1} \to L^{q_2}} \leq C_{\mathcal{L}} t^{-\nu(\frac{1}{q_1} - \frac{1}{q_2})}, \quad t \in (0, 1], \tag{1}$$

*for some $\nu > 0$.*

**Assumption 2.** *$F \in C^1(\mathbb{R}; \mathbb{R})$ satisfies $F(0) = 0$ and*

$$|F(z_1) - F(z_2)| \leq C_F \max_{i=1,2} |z_i|^{p-1} |z_1 - z_2|, \tag{2}$$

*for any $z_1, z_2 \in \mathbb{R}$ and for some $p > 1$ and $C_F > 0$.*

**Remark 1.** *The range $(0, 1]$ of $t$ in (1) can be generalized to $(0, T_{\mathcal{L}}]$, but it is assumed here to be $(0, 1]$ for simplicity. This generalization is not essential, as the existence time $T$ of the solution is sufficiently small in the fixed point argument later. Long time solutions are achieved by repeatedly using the solution operator of (P') constructed in the fixed point argument (see also Subsection 4.2).*

**Remark 2.** *Typical examples of $\mathcal{L}$ are the Laplacian with the Dirichlet, Neumann, or Robin boundary condition, the Schrödinger operator, the elliptic operator, and the higher-order Laplacian. See Appendix A for the details. On the other hand, typical examples of the nonlinearity $F$ are $F(u) = \pm|u|^{p-1}u, \pm|u|^p$ (which can be regarded as the main term of the Taylor expansion of a more general nonlinearity $F$ if $F$ is smooth in some extent). See Appendix G for further remarks on Assumptions 1 and 2.*

Under the above assumptions, the problem (P') is local well-posed, where, as a solution space, we use the space $L^r(0, T; L^s(D))$ with the parameters $r, s$ satisfying

$$r, s \in [p, \infty] \quad \text{and} \quad \frac{\nu}{s} + \frac{1}{r} < \frac{1}{p-1}. \tag{3}$$

More precisely, we have the following result on LWP.

**Proposition 1.** *Assume that $r, s$ satisfy (3). Then, for any $u_0 \in L^\infty(D)$, there exist a time $T = T(u_0) \in (0, 1]$ and a unique solution $u \in L^r(0, T; L^s(D))$ to (P'). Moreover, for any $u_0, v_0 \in L^\infty(D)$, the solutions $u$ and $v$ to (P') with $u(0) = u_0$ and $v(0) = v_0$ satisfy the continuous dependence on initial data: There exists a constant $C > 0$ such that*

$$\|u - v\|_{L^r(0,T';L^s)} \leq C\|u_0 - v_0\|_{L^\infty},$$

*where $T' < \min\{T(u_0), T(v_0)\}$.*

The proof is based on the fixed point argument. Given an initial data $u_0 \in L^\infty(D)$ and $T, M > 0$, we define the map $\Phi = \Phi_{u_0}$ by

$$\Phi[u](t) := S_{\mathcal{L}}(t)u_0 + \int_0^t S_{\mathcal{L}}(t - \tau) F(u(\tau)) \, d\tau \tag{4}$$

for $t \in [0, T]$ and the complete metric space $X := B_{L^r(0,T;L^s(D))}(M)$ equipped with the metric

$$\mathrm{d}(u, v) := \|u - v\|_{L^r(0,T;L^s)}.$$

Let $R, M > 0$ be arbitrarily fixed, and let $T > 0$ (which is taken sufficiently small later). Then, under Assumptions 1 and 2, it can be proved that for any $u_0 \in B_{L^\infty}(R)$, the map $\Phi : X \to X$ is $\delta$-contractive with a contraction rate $\delta \in (0, 1)$, i.e.

$$\mathrm{d}(\Phi[u], \Phi[v]) \leq \delta \mathrm{d}(u, v) \quad \text{for any } u, v \in X,$$

where $T$ is taken small enough to depend on $R$, $M$ and $\delta$ (not to depend on $u_0$ itself). Therefore, Banach's fixed point theorem allows us to prove that there exists uniquely a function $u \in X$ such that $u = \Phi[u]$ (a fixed point). This function $u$ is precisely the solution of (P') with the initial data $u(0) = u_0$. Thus, Proposition 1 is shown. See Appendix B for more details of the proof.

This proof by the fixed point argument guarantees the following result (see e.g. Zeidler (1986)).

**Proposition 2.** *Assume that $r, s$ satisfy (3). Then, for any $R, M > 0$ and for any $\delta \in (0, 1)$, there exists a time $T \in (0, 1]$, depending on $R, M$ and $\delta$, such that the following statements hold:*

(i) *There exists a unique solution operator $\Gamma^+ : B_{L^\infty}(R) \to B_{L^r(0,T;L^s)}(M)$ such that*

$$\Gamma^+(u_0) = u$$

*for any $u_0 \in B_{L^\infty}(R)$, where $u$ is the solution to (P') with $u(0) = u_0$ given in Proposition 1.*

(ii) *Given $u_0 \in B_{L^\infty}(R)$, define Picard's iteration by*

$$\begin{cases} u^{(1)} := S_{\mathcal{L}}(t)u_0, \\ u^{(\ell+1)} := \Phi[u^{(\ell)}] = u^{(1)} + \displaystyle\int_0^t S_{\mathcal{L}}(t - \tau)F(u^{(\ell)}(\tau)) \, d\tau, \quad \ell = 1, 2, \cdots, \end{cases} \quad (5)$$

*that is,*

$$u^{(\ell)} := \Phi^{[\ell]}[0] = \underbrace{\Phi \circ \cdots \circ \Phi}_{\ell \text{ times}}[0], \quad \ell = 1, 2, \cdots,$$

*where $\Phi : X \to X$ is a $\delta$-contraction mapping defined by (4). Then $u^{(\ell)} \to u$ in $L^r(0, T; L^s(D))$ as $\ell \to \infty$ and*

$$\mathrm{d}(u^{(\ell)}, u) \leq \frac{\delta^\ell}{1 - \delta}\mathrm{d}(u^{(1)}, 0), \quad \ell = 1, 2, \cdots.$$

## 3 NEURAL OPERATOR FOR NONLINEAR PARABOLIC EQUATIONS

In this section, we aim to construct neural operators $\Gamma$ that serve as accurate approximation models of the solution operators $\Gamma^+$ for nonlinear parabolic PDEs (P). We start by explaining our idea in rough form. Our idea is inspired by the fixed point argument and Picard's iteration. By Section 2, for any $u_0 \in B_{L^\infty}(R)$, the solution $u$ to (P') on $[0, T] \times D$ can be obtained through the Picard's iteration (5) under appropriate settings. If the semigroup $S_{\mathcal{L}}(t)$ has an integral kernel $G = G(t, x, y)$, which is the Green function $G$ of the linear equation $\partial_t u + \mathcal{L}u = 0$, then we can write

$$S_{\mathcal{L}}(t)u_0(x) = \int_D G(t, x, y)u_0(y) \, dy,$$

$$\int_0^t S_{\mathcal{L}}(t - \tau)F(u^{(\ell)}(\tau, x)) \, d\tau = \int_0^t \int_D G(t - \tau, x, y)F(u^{(\ell)}(\tau, y)) \, dy d\tau.$$

Suppose that $G$ has an expansion

$$G(t - \tau, x, y) = \sum_{m,n \in \Lambda} c_{m,n}\psi_m(\tau, y)\varphi_n(t, x) = \lim_{N \to \infty} \sum_{m,n \in \Lambda_N} c_{m,n}\psi_m(\tau, y)\varphi_n(t, x),$$

for $0 \leq \tau, t \leq T$ and $x, y \in D$. For convenience, we always assume that $G(t, x, y) = 0$ for $t \leq 0$. Here, $\Lambda$ is an index set that is either finite or countably infinite, and $\Lambda_N$ is a subset of $\Lambda$ with its cardinality $|\Lambda_N| = N \in \mathbb{N}$ and the monotonicity $\Lambda_N \subset \Lambda_{N'}$ for any $N \leq N'$. We write the partial sum as

$$G_N(t - \tau, x, y) := \sum_{m,n \in \Lambda_N} c_{m,n}\psi_m(\tau, y)\varphi_n(t, x)$$

for $0 \leq \tau, t \leq T$ and $x, y \in D$. We always assume that $G_N(t, x, y) = 0$ for $t \leq 0$ as well. We define $\Phi_N$ by

$$\begin{aligned} \Phi_N[u](t, x) &:= \int_D G_N(t, x, y)u_0(y) \, dy + \int_0^t \int_D G_N(t - \tau, x, y)F(u(\tau, y)) \, dy d\tau \\ &= \sum_{m,n \in \Lambda_N} c_{m,n}\langle\psi_m(0, \cdot), u_0\rangle\varphi_n(t, x) + \sum_{m,n \in \Lambda_N} c_{m,n}\langle\psi_m, F(u)\rangle\varphi_n(t, x) \\ &= \sum_{m,n \in \Lambda_N} c_{m,n}\left(\langle\psi_m(0, \cdot), u_0\rangle + \langle\psi_m, F(u)\rangle\right)\varphi_n(t, x), \end{aligned}$$

and moreover, we define an approximate Picard's iteration by

$$\begin{cases} \hat{u}^{(1)} := \displaystyle\int_D G_N(t,x,y)u_0(y)\,dy = \sum_{m,n\in\Lambda_N} c_{m,n}\langle\psi_m(0,\cdot),u_0\rangle\varphi_n(t,x), \\ \hat{u}^{(\ell+1)} := \Phi_N[\hat{u}^{(\ell)}], \quad \ell = 1,2,\cdots. \end{cases}$$

Then, for sufficiently large $N$, we expect:

1. $\Phi_N \approx \Phi$ and $\Phi_N$ is also contractive on $X_N := B_{L^r(0,T_N;L^s)}(M)$ for some $T_N > 0$.

2. There exists a fixed point $\hat{u} \in X_N$ of $\Phi_N$ such that $\hat{u}^{(\ell)} \to \hat{u}$ as $\ell \to \infty$.

3. $\hat{u} \approx u$ on $[0, \min\{T, T_N\}] \times D$ for any $u_0 \in B_{L^\infty}(R)$.

4. Define $\Gamma : B_{L^\infty}(R) \to X_N$ by $\Gamma(u_0) := \hat{u}^{(L)}$ for $L \in \mathbb{N}$. Then $\Gamma \approx \Gamma^+$ as $N, L \to \infty$.

The above $\Gamma$ is the prototype of our neural operator, where $N$ corresponds to the rank and $L$ to the layer depth. In other words, in our neural operator, the Picard's iteration step corresponds to the forward propagation in the layer direction of the neural operator, which converges to the fixed point (i.e. the solution to (P')) as the layers get deeper for sufficiently large rank $N$.

The above is only a rough idea. The precise definition of our neural operator is the following:

**Definition 1** (Neural operator). *Let $T > 0$ and $r, s \in [1, \infty]$. Let $\varphi := \{\varphi_n\}_n$ and $\psi := \{\psi_m\}_m$ be families of functions in $L^r(0,T;L^s(D))$ and $L^{r'}(0,T;L^{s'}(D))$, respectively. We define a neural operator $\Gamma : L^\infty(D) \to L^r(0,T;L^s(D))$ by*

$$\Gamma : L^\infty(D) \to L^r(0,T;L^s(D)) : u_0 \mapsto \hat{u}^{(L+1)}.$$

*Here, the output function $\hat{u}^{(L+1)}$ is given by the following steps:*

*1. (**Input layer**) $\hat{u}^{(1)} = (\hat{u}_1^{(1)}, \hat{u}_2^{(1)}, \ldots, \hat{u}_{d_1}^{(1)})$ is given by*

$$\hat{u}^{(1)}(t,x) := (K_N^{(0)}u_0)(t,x) + b_N^{(0)}(t,x).$$

*Here, $K_N^{(0)} : L^\infty(D) \to L^r(0,T;L^s(D))^{d_1}$ and $b_N^{(0)} \in L^r(0,T;L^s(D))^{d_1}$ are defined by*

$$(K_N^{(0)}u_0)(t,x) := \sum_{m,n\in\Lambda_N} C_{n,m}^{(0)}\langle\psi_m(0,\cdot),u_0\rangle\varphi_n(t,x) \quad \text{with } C_{n,m}^{(0)} \in \mathbb{R}^{d_1\times 1},$$

$$b_N^{(0)}(t,x) := \sum_{n\in\Lambda_N} b_N^{(0)}\varphi_n(t,x) \quad \text{with } b_N^{(0)} \in \mathbb{R}^{d_1}.$$

*2. (**Hidden layers**) For $2 \le \ell \le L$, $\hat{u}^{(\ell)} = (\hat{u}_1^{(\ell)}, \hat{u}_2^{(\ell)}, \ldots, \hat{u}_{d_\ell}^{(\ell)})$ are iteratively given by*

$$\hat{u}^{(\ell+1)}(t,x) = \sigma\left(W^{(\ell)}\hat{u}^{(\ell)}(t,x) + (K_N^{(\ell)}\hat{u}^{(\ell)})(t,x) + b^{(\ell)}\right), \quad 1 \le \ell \le L-1.$$

*3. (**Output layer**) $\hat{u}^{(L+1)}$ is given by*

$$\hat{u}^{(L+1)}(t,x) = W^{(L)}\hat{u}^{(L)}(t,x) + (K_N^{(L)}\hat{u}^{(L)})(t,x) + b^{(L)}.$$

*Here, $\sigma : \mathbb{R} \to \mathbb{R}$ is a nonlinear activation operating element-wise, and $W^{(\ell)} \in \mathbb{R}^{d_{\ell+1}\times d_{\ell+1}}$ is a weight matrix of the $\ell$-th hidden layer, and $b^{(\ell)} \in \mathbb{R}^{d_\ell}$ is a bias vector, and $K_N^{(\ell)} : L^r(0,T;L^s(D))^{d_\ell} \to L^r(0,T;L^s(D))^{d_{\ell+1}}$ is defined by*

$$(K_N^{(\ell)}u)(t,x) := \sum_{m,n\in\Lambda_N} C_{n,m}^{(\ell)}\langle\psi_m,u\rangle\varphi_n(t,x) \quad \text{with } C_{n,m}^{(\ell)} \in \mathbb{R}^{d_{\ell+1}\times d_\ell},$$

*where we use the notation*

$$\langle\psi_m,u\rangle := (\langle\psi_m,u_1\rangle, \ldots, \langle\psi_m,u_{d_\ell}\rangle) \in \mathbb{R}^{d_\ell},$$

*for $u = (u_1, \ldots, u_{d_\ell}) \in L^r(0,T;L^s(D))^{d_\ell}$. Note that $d_{L+1} = 1$.*

*We denote by $\mathcal{NO}_{N,\varphi,\psi}^{L,H,\sigma}$ the class of neural operators defined as above, with the depth $L$, the number of neurons $H = \sum_{\ell=1}^L d_\ell$, the rank $N$, the activation function $\sigma$, and the families of functions $\varphi, \psi$.*

The operators $K_N^{(l)}$ play a crucial role in capturing the non-local nature of PDEs. They are defined by truncating the basis expansion, a definition for neural operators inspired by Lanthaler et al. (2023). In this context, $K_N^{(l)}$ are finite-rank operators with rank $N$. The model complexity is determined not only by the depth $L$ and the number $H$ of neurons, but also by the rank $N$. The families $\varphi$ and $\psi$ are hyperparameters, which are chosen so that their expansions can approximate the Green function. Examples are the Fourier basis, wavelet basis, orthogonal polynomial, spherical harmonics, and eigenfunctions of $\mathcal{L}$. When we select $\varphi$ and $\psi$ as the Fourier basis and wavelet basis, the resulting network architectures correspond to the Fourier neural operator (FNO) in Li et al. (2020a) and the wavelet neural operator (WNO) in Tripura & Chakraborty (2023), respectively. See also Appendix C for FNOs and WNOs in some simple cases.

## 4 QUANTITATIVE APPROXIMATION THEOREM

In this section we prove a quantitative approximation theorem for our neural operator $\Gamma$ (Definition 1). For this purpose, we assume that $\mathcal{L}$ and $F$ satisfy Assumptions 1 and 2, respectively, and the parameters $r, s$ satisfy the condition (3). In addition, we assume the following:

**Assumption 3.** *Let $\Lambda$ be an index set that is either finite or countably infinite and $\Lambda_N$ a subset of $\Lambda$ with its cardinality $|\Lambda_N| = N \in \mathbb{N}$ and the monotonicity $\Lambda_N \subset \Lambda_{N'}$ for any $N \leq N'$. For the Green function $G$ of the linear equation $\partial_t u + \mathcal{L}u = 0$, there exist the families of functions $\varphi := \{\varphi_n\}_{n \in \Lambda}$ and $\psi := \{\psi_m\}_{m \in \Lambda}$ in $L^r(0, T; L^s(D))$ and $L^{r'}(0, T; L^{s'}(D))$, respectively, such that $G$ has the following expansions:*

$$G(t - \tau, x, y) = \sum_{m,n \in \Lambda} c_{n,m} \psi_m(\tau, y) \varphi_n(t, x), \quad 0 < \tau, t < T, \ x, y \in D, \tag{6}$$

$$G(t, x, y) = \sum_{m,n \in \Lambda} c_{n,m} \psi_m(0, y) \varphi_n(t, x), \quad 0 < t < T, \ x, y \in D \tag{7}$$

*in the sense that*

$$E_G(N) := \left\| \|G(t - \tau, x, y) - G_N(t - \tau, x, y)\|_{L_\tau^{r'}(0,T;L_y^{s'})} \right\|_{L_t^r(0,T;L_x^s)} \to 0,$$

$$E'_G(N) := \left\| \|G(t, x, y) - G_N(t, x, y)\|_{L_y^{s'}} \right\|_{L_t^r(0,T;L_x^s)} \to 0$$

*as $N \to \infty$, respectively, where*

$$G_N(t - \tau, x, y) := \sum_{m,n \in \Lambda_N} c_{n,m} \psi_m(\tau, y) \varphi_n(t, x), \quad 0 \leq \tau, t < T, \ x, y \in D. \tag{8}$$

### 4.1 MAIN RESULT

Our main result is the following quantitative approximation theorem.

**Theorem 1.** *Suppose that $\mathcal{L}$ and $F$ satisfy Assumptions 1 and 2, respectively, and the parameters $r, s$ satisfy the condition (3). Then for any $R > 0$, there exists a time $T > 0$ such that the following statement holds under Assumption 3: For any $\epsilon \in (0, 1)$, there exist a depth $L$, the number of neurons $H$, a rank $N$, and $\Gamma \in \mathcal{NO}_{N,\varphi,\psi}^{L,H,ReLU}$ such that*

$$\sup_{u_0 \in B_{L^\infty}(R)} \|\Gamma^+(u_0) - \Gamma(u_0)\|_{L^r(0,T;L^s)} \leq \epsilon.$$

*Moreover, $L = L(\Gamma)$ and $H = H(\Gamma)$ satisfy*

$$L(\Gamma) \leq C_L (\log(\epsilon^{-1}))^2 \quad and \quad H(\Gamma) \leq C_H \epsilon^{-1} (\log(\epsilon^{-1}))^2$$

*for some positive constants $C_L$ and $C_H$ depending on $\nu, M, F, D, p, r, s, R$, and $\mathcal{L}$, and $N = N(\Gamma)$ satisfies*

$$E_G(N(\Gamma)) \leq C_G \epsilon \quad and \quad E'_G(N(\Gamma)) \leq C_G \epsilon$$

*for some positive constant $C_G$ depending on $\nu, M, F, D, p, r, s, R$, and $\mathcal{L}$.*

Let us give a remark on model complexity of our neural operator $\Gamma$ in Theorem 1. The model complexity depends on the depth $L(\Gamma)$, the number of neurons $H(\Gamma)$, and the rank $N$. We observe that $L(\Gamma)$ increases only squared logarithmically at most as $\epsilon \to 0$ and $H(\Gamma)$ increases on the order $\epsilon^{-1}$ at most as $\epsilon \to 0$. This is reasonable since each forward pass of the neural operator's layers corresponds to a step in Picard's iteration, and the convergence rate of Picard's iteration decays exponentially (see Proposition 2 (ii)). As to the rank $N$, Theorem 1 does not mention anything regarding the rates of the errors $E_G(N)$ and $E'_G(N)$, but these error rates are also necessary for quantitatively estimating the model complexity of $\Gamma$. These error rates strongly depend on the choice of $\varphi$ and $\psi$, and the research results are enormous (see e.g. Cohen (2003); DeVore & Popov (1988); DeVore et al. (1993); Dũng (2011)). For example, in the case of Fourier bases on $d$-dimensional torus $\mathbb{T}^d$, it can be shown by Parseval's identity that the approximation rate in $L^2$-norm is $O(N^{-\sigma+d/2})$ $(\sigma > d/2)$ if the target function belongs to the $L^2$ type Sobolev space $H^\sigma(\mathbb{T}^d)$ (see Dyachenko (1995) for $L^p$-framework). As another example, for many wavelet bases, the approximation rate $O(N^{-\sigma/d})$ is known based on Jackson-type estimates (see Cohen (2003, Theorem 4.3.2)). Furthermore, there are cases where the rates differ between linear approximation and nonlinear approximation (wavelet approximation); in particular, when the target function is discountinuous (see Cohen (2003, Remark 4.2.5)). Therefore, if the Green function $G$ has a certain regularity, we may obtain a quantitative estimate for the rank $N$ such as the polynomial rate of $E_G(N)$ and $E'_G(N)$, and as a result, it is shown that the model complexity of $\Gamma$ does not grow exponentially. This prevents the exponential growth in model complexity, which often arises in general operator approximation.

Next, we remark that the constants $C_L$, $C_H$ and $C_G$ in Theorem 1 depend on $D$, in particular on the dimension $d$. According to the proof of Theorem 1 (especially, Lemma 4 in Appendix D), the dependencies on $D$ are $C_L \approx 1$, $C_H \approx |D|^{1/s}$ and $C_G \lesssim 1$ for sufficiently small $\epsilon > 0$. For simplicity, letting $D = [0, R]^d$, we see that the growth of $C_H$ with respect to $d$ is $C_H \approx R^{d/s}$.

*Sketch of proof.* (See Appendix D for the exact proof). The proof is based on the following scheme:

$$u_0 \xrightarrow{S_\mathcal{L}} u^{(1)} \xrightarrow{\Phi} u^{(2)} \xrightarrow{\Phi} \cdots \xrightarrow{\Phi} u^{(J)}(\cdots \xrightarrow{J \to \infty} u) \qquad \text{(Picard's iteration)}$$

$$u_0 \xrightarrow{S_{\mathcal{L},N}} \hat{u}^{(1)} \xrightarrow{\Phi_{N,net}} \hat{u}^{(2)} \xrightarrow{\Phi_{N,net}} \cdots \xrightarrow{\Phi_{N,net}} \hat{u}^{(J)} \qquad \text{(Neural operator iteration)}$$

where $S_\mathcal{L}$, $S_{\mathcal{L},N}$, $\Phi$, and $\Phi_{N,net}$ are defined as

$$S_\mathcal{L}[u_0](t, x) := \int_D G(t, x, y) u_0(y)\, dy, \quad S_{\mathcal{L},N}[u_0](t, x) := \int_D G_N(t, x, y) u_0(y)\, dy,$$

$$\Phi[u](t, x) := S_\mathcal{L}[u_0](t, x) + \int_0^t \int_D G(t, x, y) F(u(\tau, y))\, dy d\tau,$$

$$\Phi_{N,net}[u](t, x) := S_{\mathcal{L},N}[u_0](t, x) + \int_0^t \int_D G_N(t, x, y) F_{net}(u(\tau, y))\, dy d\tau.$$

First, by the truncation of Picard's iteration, the solution $u = \Gamma^+(u_0)$ to (P') is approximated by

$$u \approx u^{(J)} = \Phi^{[J-1]} \circ S_\mathcal{L}[u_0],$$

for sufficiently large $J$. Next, by approximating the Green function $G$ and the nonlinearity $F$ with a truncated expansion $G_N$ and a neural network $F_{net}$, respectively, the contraction mapping $\Phi$ is approximated by $\Phi_{N,net}$ for sufficiently large $N$ (Lemmas 2 and 3), leading to

$$u^{(J)} \approx \hat{u}^{(J)} = \Phi_{N,net}^{[J-1]} \circ S_{\mathcal{L},N}[u_0],$$

(Lemma 4). Finally, we represent $\hat{u}^{(J)}$ as a neural operator $\Gamma$ (Lemma 5).

The key idea is to mimic the Picard's iteration. The convergence rate of Picard's iteration is given in Proposition 2 (ii), which reads $J \approx \log(\epsilon^{-1})$. By applying the one dimensional case of Gühring et al. (2020, Theorem 4.1), we approximate $F \approx F_{net}$ with the rates $L(F_{net}) \lesssim (\log(\epsilon^{-1}))$ and $H(F_{net}) \lesssim \epsilon^{-1}(\log(\epsilon^{-1}))$. Note that the curse of dimensionality that occurs in conventional neural network approximations does not appear here. Consequently, the rates of the depth $L(\Gamma)$ and the number of neurons $H(\Gamma)$ are evaluated as

$$L(\Gamma) \lesssim L(F_{net}) \cdot J \approx \log\left(\epsilon^{-1}\right) \cdot \log(\epsilon^{-1}) = (\log(\epsilon^{-1}))^2,$$

$$H(\Gamma) \lesssim H(F_{net}) \cdot J \lesssim \epsilon^{-1}\left(\log(\epsilon^{-1})\right) \cdot \left(\log(\epsilon^{-1})\right) \lesssim \epsilon^{-1}\left(\log(\epsilon^{-1})\right)^2,$$

where the first inequality is given by the exact construction of the neural operator $\Gamma$ discussed in the proof of Lemma 5 as

$$\Gamma(u_0) = \tilde{W}' \circ \underbrace{\left[ (\tilde{W} + \tilde{K}_N) \circ \tilde{F}_{net} \circ \cdots \circ (\tilde{W} + \tilde{K}_N) \circ \tilde{F}_{net} \right]}_{J-1} \circ (\tilde{K}_N^{(0)} + \tilde{b}_N^{(0)})(u_0), \quad (9)$$

where $\tilde{W}'$ and $\tilde{W}$ are some weight matrices, $\tilde{K}_N$ and $\tilde{K}_N^{(0)}$ are some non-local operators, $\tilde{b}_N^{(0)}$ is some bias function, and $\tilde{F}_{net}$ is some neural network so that $L(\tilde{F}_{net}) = L(F_{net})$. Note that the operation $(\tilde{W} + \tilde{K}_N) \circ \tilde{F}_{net}$ corresponds to the one operation $\Phi_{N,net}$, and it can be a $L(F_{net}) + 1$ layer neural operator. In this construction, when denoting by $(\tilde{W} + \tilde{K}_N) \circ \tilde{F}_{net} = (W^{(L(F_{net})+1)} + K^{(L(F_{net})+1)} + b^{(L(F_{net})+1)}) \circ \sigma \circ \cdots \circ \sigma \circ (W^{(1)} + K^{(1)} + b^{(1)})$, the local operators $W^{(\ell)}$ and biases $b^{(\ell)}$ for $\ell = 1, \ldots, L(F_{net})$ correspond to those in the neural network $\tilde{F}_{net}$, while the non-local operators $K^{(\ell)} \equiv 0$ for $\ell = 1, \ldots, L(F_{net})$. At the $L(F_{net}) + 1$-th layer, $W^{(L(F_{net})+1)} = \tilde{W}$ and $b^{(L(F_{net})+1)} \equiv 0$, and the non-local operator $K^{(L+1)} = \tilde{K}_N$. $\qquad\square$

**Remark 3.** *It is important to note that the hidden layers of the neural operator $\Gamma$ constructed in (9) are structured to repeatedly apply the same operator, $(\tilde{W} + \tilde{K}_N) \circ \tilde{F}_{net}$, which significantly reduces memory consumption. Our constructed neural operator is closely related to the deep equilibrium models (Bai et al., 2019), which utilize weight-tied architectures to find fixed points. More recently, inspired by deep equilibrium models, weight-tied neural operators have been experimentally investigated in the context of solving steady-state PDEs (Marwah et al., 2023). Based on the above findings, our results also give weight-tied neural operators for (unsteady-state) parabolic PDEs and theoretically guarantee their approximation ability.*

**Remark 4.** *One of the next issues in approximation theory is to verify in what sense the obtained approximate solution is a good approximation. In particular, it is important to discuss that an approximator $\Gamma$ obtained in Theorem 1 preserves desired properties in parabolic PDEs, such as the maximum principle and the comparison principle. In the special case of the nonlinearity, we can show that approximator $\Gamma$ preserves the positivity. For more details, see Appendix F.*

## 4.2 APPROXIMATION OF LONG TIME SOLUTIONS

The problem of approximating global in time solutions using neural operators is a more challenging and significant task. It is known that global solutions can be extended by repeatedly applying the solution operator $\Gamma^+$ when the $L^\infty$-norm of the solution remains uniformly bounded. Based on this idea, this section discusses the approximation of long time solutions using neural operators.

**Extension of a short time solution to a long time solution.** By Proposition 2 (i), for any initial data $u_0 \in B_{L^\infty}(R)$, there exists a short time solution $u = \Gamma^+(u_0)$ to (P') on $[0, T] \times D$. Then, if $u(T)$ at $t = T$ belongs to $B_{L^\infty}(R)$, the same solution operator $\Gamma^+$ can be applied to $u(T)$ as an initial data, and hence, the solution can be extended to that on $[0, 2T] \times D$. Thus, as long as the supremum value of the solution is less than or equal to $R$, the solution can be extended with respect to $t$ by applying $\Gamma^+$ as many times as possible. We denote by $u_{\max}$ the extended maximal solution with respect to $t$ and by $\kappa_{\max} = \kappa_{\max}(u_0) \in \mathbb{N} \cup \{\infty\}$ the maximal number of extensions by $\Gamma^+$. Then we can write $u_{\max}$ as

$$u_{\max}(t) = \Gamma^{+[\kappa]}(u_0)(t)$$

for $(\kappa - 1)T < t \le \kappa T$ and $\kappa = 1, \ldots, \kappa_{\max}$.

**Approximation of the long time solution $u_{\max}$ by our neural operator.** Similarly, for our neural operator $\Gamma$ given in Theorem 1, the output function of $\Gamma$ can be extended with respect to time $t$ by applying $\Gamma$ as many times as possible. We denote by $\hat{u}_{\max}$ the extended maximal output function of $\Gamma$ with respect to $t$ and by $\hat{\kappa}_{\max} = \hat{\kappa}_{\max}(u_0) \in \mathbb{N} \cup \{\infty\}$ the maximal number of extensions by $\Gamma$. Then we can also write $\hat{u}_{\max}$ as

$$\hat{u}_{\max}(t) := \Gamma^{[\kappa]}(u_0)(t)$$

for $(\kappa - 1)T < t \le \kappa T$ and $\kappa = 1, \ldots, \hat{\kappa}_{\max}$. Then the error between $u_{\max}$ and $\hat{u}_{\max}$ can be controlled by the error $\epsilon$ in Theorem 1 and another error $\tilde{\epsilon}$ given by

$$\tilde{\epsilon} := \sum_{\kappa=1}^{\kappa^*-1} \|\Gamma^{+[\kappa]}(u_0)(\kappa T) - \Gamma^{[\kappa]}(u_0)(\kappa T)\|_{L^\infty}.$$

More precisely, we have the following result on the error between $u_{\max}$ and $\hat{u}_{\max}$.

**Corollary 1.** *Let* $\epsilon \in (0, 1)$ *and* $R > 0$. *Suppose the same assumptions as in Theorem 1 and* $\kappa^* := \min\{\kappa_{\max}, \hat{\kappa}_{\max}\}$ *is a finite number. Then there exists a constant* $C > 0$ *such that*

$$\|u_{\max} - \hat{u}_{\max}\|_{L^r(0,\kappa^* T; L^s)} \leq C(\epsilon + \tilde{\epsilon})$$

*for any* $u_0 \in B_{L^\infty}(R)$, *where* $T$ *and* $\Gamma$ *are the same as in Theorem 1.*

The proof is given in Appendix E.

## 5 DISCUSSION AND FUTURE WORK

**Applicability and limitation.** In this paper we focus on parabolic PDEs with power-type (or somewhat general) nonlinear terms as a first step of study of neural operators based on Picard's iteration. However, we believe that our approach is not restricted to parabolic PDEs and can be adapted to a broader class of nonlinear PDEs, because the Banach's fixed point theory is widely applicable to many nonlinear PDEs (see Appendix G for the details). On the other hand, for PDEs that involve nonlinearity in the principal part such as $p$-Laplacian equations $\partial_t u - \Delta_p u = F(u)$ with $\Delta_p := \mathrm{div}(|\nabla u|^{p-2}\nabla u)$ and $p \neq 2$, it may not be possible to represent them in the form of integral equations. Such equations might not be effectively handled by our approach.

**Further development of Theorem 1.** In this paper we discuss the LWP for (P) and the approximation error in Theorem 1 in the framework of the Lebesgue spaces. However, we expect that these results can be generalized to those in the framework of Sobolev spaces. In fact, LWP can be established in the Sobolev space framework as shown in Ribaud (1998). Further, it is also possible to discuss the approximation error in the same framework, since the basis expansion of the Green function and the approximation of nonlinearity by neural networks are studied in Sobolev spaces.

Next, in the results of this paper, only bounded domains $D$ are considered due to technical reasons in the proofs. However, the question of whether Theorem 1 can be extended to general domains, including unbounded ones, is an interesting problem (at least mathematically) and remains a direction for future work.

Moreover, in this paper we discuss the problem in the setting of an abstract operator $\mathcal{L}$ and general bases $\varphi, \psi$. However, it is also important to develop a more detailed analysis by restricting the setting to more specific operators and basis functions. For example, by focusing on specific operators and basis functions, one can construct approximation theorems that include concrete rates for rank $N$, and it is crucial to investigate how these rates vary depending on the choice of operators and basis functions. This issue remains one of the directions for future work.

**Implementation.** This paper does not address the implementation of neural operators, so we briefly comment it here. Our proof based on Picard's iteration is constructive, and this constructive approach may offer valuable insights for future experimental studies, particularly when incorporating constraints specific to the architecture of PDE tasks (for instance, weight-tied architecture discussed in Remark 3). Note that, due to the form of integral equation (P'), our neural operators need to include the integral with respect to time, which may be computationally expensive. However, the previous work Kovachki et al. (2023, Section 7.3) employed the Fourier transforms with respect to time for computing FNOs, whose techniques might be useful for computing our neural operators. We leave the details of experimental studies of our theory in the future works.

**Neural operators with desirable properties.** In numerical analysis, it is crucial that the solutions of discrete equations retain the properties of the original continuous solutions. Similarly, neural operators must also preserve these properties, especially when used as surrogate models of simulators in practical applications. While we have shown positivity preserving in a specific case of nonlinearity (Remark 4), future work may extend this result to more general nonlinear settings. Additionally, other important properties, such as the comparison principle and the smoothing effect, which are essential in parabolic equations, will be the focus of future studies.

ACKNOWLEDGMENTS

T. Furuya is supported by JSPS KAKENHI Grant Number JP24K16949, JST CREST JP-MJCR24Q5, and JST ASPIRE JPMJAP2329. K. Taniguchi is supported by JSPS KAKENHI Grant Number 24K21316.

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

APPENDIX

# A    EXAMPLES OF $\mathcal{L}$ (ASSUMPTION 1)

This appendix provides some typical examples of $\mathcal{L}$ satisfying Assumption 1.

## A.1    DIRICHLET LAPLACIAN

Let $D$ be a bounded domain in $\mathbb{R}^d$ with $d \in \mathbb{N}$. Let $\mathcal{L} = -\Delta_{\mathrm{Dir}}$ denote the Dirichlet Laplacian on $L^2(D)$. Then the semigroup $\{S_{-\Delta_{\mathrm{Dir}}}(t)\}_{t \geq 0}$ corresponds to the solution operator of the initial boundary value problem of the linear heat equation

$$
\begin{cases}
\partial_t u - \Delta u = 0 & \text{in } (0,T) \times D, \\
u(0) = u_0 & \text{in } D, \\
u = 0 & \text{on } (0,T) \times \partial D,
\end{cases}
$$

where $\partial D$ denotes the boundary of $D$. The integral kernel $G_{\mathrm{Dir}}(t,x,y)$ of $S_{-\Delta_{\mathrm{Dir}}}(t)$ has the Gaussian upper bound

$$
0 \leq G_{\mathrm{Dir}}(t,x,y) \leq (4\pi t)^{-\frac{d}{2}} \exp\left(-\frac{|x-y|^2}{4t}\right) =: \mathcal{G}(t, x-y),
$$

for any $t > 0$ and almost everywhere $x, y \in D$ (see e.g. Iwabuchi et al. (2018, Proposition 3.1)). Then, by combining this bound with the Young inequality, we have

$$
\begin{aligned}
\|S_{-\Delta_{\mathrm{Dir}}}(t)f\|_{L^{q_2}(D)} &\leq \left\| \int_{\mathbb{R}^d} \mathcal{G}(t, x-y) |\tilde{f}(y)| \, dy \right\|_{L^{q_2}(\mathbb{R}^d)} \\
&\leq C t^{-\frac{d}{2}\left(\frac{1}{q_1} - \frac{1}{q_2}\right)} \|\tilde{f}\|_{L^{q_1}(\mathbb{R}^d)} = C t^{-\frac{d}{2}\left(\frac{1}{q_1} - \frac{1}{q_2}\right)} \|f\|_{L^{q_1}(D)}
\end{aligned}
$$

for any $f \in L^{q_1}(D)$, where $1 \leq q_1 \leq q_2 \leq \infty$ and $\tilde{f}$ is the zero extension of $f$ to $\mathbb{R}^d$. Therefore, we see that $\mathcal{L} = -\Delta_{\mathrm{Dir}}$ satisfies Assumption 1 with $\nu = d/2$.

## A.2    NEUMANN OR ROBIN LAPLACIAN

Let $D$ be a bounded Lipschitz domain in $\mathbb{R}^d$ with $d \geq 1$. We consider the Robin Laplace $-\Delta_\gamma$ on $L^2(D)$ associated with a quadratic form

$$
q_\gamma(f, g) = \int_\Omega \nabla f \cdot \nabla g \, dx + \int_{\partial \Omega} \gamma f g \, dS,
$$

for any $f, g \in H^1(D)$, where $\gamma$ is a function $\partial\Omega \to \mathbb{R}$. Note that $\gamma$ means a Robin boundary condition, and particularly, the case of $\gamma = 0$ means the zero Neumann boundary condition. Hence, $-\Delta_0$ is the Neumann Laplacian $-\Delta_{\mathrm{Neu}}$ on $L^2(D)$. Assume that $\gamma \in L^\infty(\partial D)$ and $\gamma \geq 0$. We denote by $G_\gamma(t,x,y)$ and $G_{\mathrm{Neu}}(t,x,y)$ the integral kernels of semigroups generated by $-\Delta_\gamma$ and $-\Delta_{\mathrm{Neu}}$, respectively. Then $G_\gamma(t,x,y)$ and $G_{\mathrm{Neu}}(t,x,y)$ have the Gaussian upper bounds:

$$
G_{\mathrm{Dir}}(t,x,y) \leq G_\gamma(t,x,y) \leq G_{\mathrm{Neu}}(t,x,y) \leq C t^{-\frac{d}{2}} \exp\left(-\frac{|x-y|^2}{C't}\right),
$$

for any $0 < t \leq 1$ and almost everywhere $x, y \in D$ and for some $C, C' > 0$. The first and second inequalities follow from domination of semigroups (see e.g. Ouhabaz (2005, Theorem 2.24)), and the proof of the last inequality can be found in Choulli et al. (2015). These bounds yield that $-\Delta_{\mathrm{Neu}}$ and $-\Delta_\gamma$ satisfy Assumption 1 with $\nu = d/2$ in the same way as Appendix A.1.

## A.3    SCHRÖDINGER OPERATOR WITH A SINGULAR POTENTIAL

Let $D$ be a bounded domain in $\mathbb{R}^d$ with $d \geq 1$. We consider the Schrödinger operator $\mathcal{L} = -\Delta + V$ with the zero Dirichlet boundary condition on $\partial D$, where $V = V(x)$ is a real-valued measurable function on $D$ such that

$$
V = V_+ - V_-, \quad V_\pm \geq 0, \quad V_+ \in L^1_{\mathrm{loc}}(D) \quad \text{and} \quad V_- \in K_d(D).
$$

We say that $V_-$ belongs to the Kato class $K_d(D)$ if

$$\begin{cases} \lim_{r \to 0} \sup_{x \in D} \int_{D \cap \{|x-y|<r\}} \dfrac{V_-(y)}{|x-y|^{d-2}} \, dy = 0 & \text{for } d \geq 3, \\[2mm] \lim_{r \to 0} \sup_{x \in D} \int_{D \cap \{|x-y|<r\}} \log(|x-y|^{-1}) V_-(y) \, dy = 0 & \text{for } d = 2, , \\[2mm] \sup_{x \in D} \int_{D \cap \{|x-y|<1\}} V_-(y) \, dy < \infty & \text{for } d = 1 \end{cases}$$

(see Simon (1982, Section A.2)). It is readily seen that the positive part $V_+$ is almost unrestricted, and the negative part $V_-$ allows for the singularity such as $V_-(x) = 1/|x|^\alpha$ with $0 \leq \alpha < 2$ if $d \geq 2$ and $0 \leq \alpha < 1$ if $d = 1$. Then it is known that the integral kernel $G_V$ of semigroup generated by the Schrödinger operator $\mathcal{L} = -\Delta + V$ satisfies the Gaussian upper bound

$$|G_V(t,x,y)| \leq Ct^{-\frac{d}{2}} \exp\left(-\frac{|x-y|^2}{C't}\right), \tag{10}$$

for any $0 < t \leq 1$ and almost everywhere $x, y \in D$ and for some $C, C' > 0$ (see e.g. Iwabuchi et al. (2018, Proposition 3.1)). Therefore, $\mathcal{L} = -\Delta + V$ satisfies Assumption 1 with $\nu = d/2$ in the same way as Appendix A.1.

## A.4 Uniformlly Elliptic Operator

We consider the following linear parabolic PDEs:

$$\begin{cases} \partial_t u + \mathcal{L}u = 0 & \text{in } (0,T) \times D, \\ u(0) = u_0 & \text{in } D, \end{cases} \tag{11}$$

where

$$\mathcal{L} := \sum_{k,j=1}^{d} \partial_{x_k}\left(a_{kj}\partial_{x_j}\right) + \sum_{k=1}^{d} b_k \partial_{x_k} + c_0,$$

with real-valued functions $a_{k,j}, b_k, c_0$ on $D$ and with domain $\mathcal{D}(\mathcal{L})$ that is a closed subspace of $H^1(D)$. Note that the choice of domain $\mathcal{D}(\mathcal{L})$ corresponds to the choice of boundary condition. For example, $\mathcal{D}(\mathcal{L}) = H_0^1(D)$ and $H^1(D)$ mean the zero Dirichlet boundary condition and the zero Neumann boundary condition, respectively. We assume that $\mathcal{L}$ satisfies the uniformly elliptic condition:

- $a_{k,j}, b_k, c_0 \in L^\infty(D)$ for all $1 \leq k, j \leq d$;
- There exists a constant $C > 0$ such that

$$\sum_{k,j=1}^{d} a_{k,j}(x)\xi_k\xi_j \geq C|\xi|^2,$$

for any $\xi \in \mathbb{R}^d$ and almost everywhere $x \in D$.

In addition, we also assume the following on $\mathcal{D}(\mathcal{L})$:

- $V$ is continuous embedded into $L^{2^*}(D)$, where $2^* = \frac{2d}{d-2}$ if $d \geq 3$, $2^*$ is any number in $(2, \infty)$ if $d = 2$, and $2^* = \infty$ if $d = 1$;
- If $u \in \mathcal{D}(\mathcal{L})$, then $\max\{0, u\}, \min\{1, |u|\} \operatorname{sign} u \in \mathcal{D}(\mathcal{L})$;
- If $u \in \mathcal{D}(\mathcal{L})$, then $e^\eta u \in \mathcal{D}(\mathcal{L})$ for any real-valued function $\eta \in C^\infty(D)$ such that $\eta$ and $|\nabla \eta|$ are bounded on $D$.

Then it is known that the integral kernel of semigroup generated by $\mathcal{L}$ satisfies the same Gaussian upper bound as (10) for any $0 < t \leq 1$ and almost everywhere $x, y \in D$ (see Ouhabaz (2005, Theorem 6.10)). Therefore, the uniformly elliptic operator $\mathcal{L}$ satisfies Assumption 1 with $\nu = d/2$.

## A.5 Other examples

Various other operators, such as the Laplacian on the $d$-dimensional torus (periodic boundary condition), the fractional Laplacian $(-\Delta)^{\alpha/2}$ ($\nu = d/\alpha$), the higher-order elliptic operator ($\nu = d/m$ with the highest order $m$) and the Schrödinger operator with a Dirac delta potential ($\nu = 1/2$ with $d = 1$), are possible, but these will not be mentioned here. For more information, see e.g. the books (Davies, 1989; Ouhabaz, 2005) and the papers (Bui et al., 2020; Davies, 1995; Ikeda et al., 2024; Iwabuchi, 2017) and references therein.

## B   Proof of Proposition 1

Let $D$ be a bounded domain in $\mathbb{R}^d$ with $d \in \mathbb{N}$. We consider the Cauchy problem of nonlinear parabolic PDEs

$$\begin{cases} \partial_t u + \mathcal{L}u = F(u) & \text{in } (0, T) \times D, \\ u(0) = u_0 & \text{in } D. \end{cases} \tag{P}$$

This problem can be formally rewritten as the integral form

$$u(t) = S_{\mathcal{L}}(t)u_0 + \int_0^t S_{\mathcal{L}}(t - \tau)F(u(\tau))\,d\tau. \tag{P'}$$

Throughout this appendix, we always assume that $\mathcal{L}$ and $F$ satisfy Assumptions 1 and 2, respectively. We study local well-posedness (LWP) for the integral equation (P'). The LWP for (P') means the existence of local in time solution, uniqueness of the solution, and continuous dependence on initial data for (P'), and it has long been extensively studied (cf. Brezis & Cazenave (1996); Weissler (1979; 1980)). In this appendix, we provide the LWP results that can be used as the basis of our study of neural operator and its approximation theorem. We use the space $L^r(0, T; L^s(D))$ as a solution space. For convenience, we set

$$\alpha = \alpha(\nu, p, s) := -\frac{\nu(p-1)}{s}, \quad \beta = \beta(p, r) := \frac{r - p + 1}{r}. \tag{12}$$

We have the following result.

**Proposition 3.** *Let $d \in \mathbb{N}$, $p \in (1, \infty)$ and $\nu > 0$. Assume that $r, s$ satisfy*

$$r, s \in [p, \infty] \quad \text{and} \quad \frac{\nu}{s} + \frac{1}{r} < \frac{1}{p - 1}. \tag{13}$$

*Let $M > 0$, $0 < T \leq 1$ and $u_0 \in L^\infty(D)$ be such that*

$$\rho + \delta M \leq M, \tag{14}$$

*where $\rho$ and $\delta$ are defined by*

$$\rho = \rho(T, \|u_0\|_{L^\infty}) := C_{\mathcal{L}}|D|^{\frac{1}{s}}T^{\frac{1}{r}}\|u_0\|_{L^\infty},$$

$$\delta = \delta(T, M) := 2(\alpha\beta^{-1} + 1)^{-\beta}C_{\mathcal{L}}C_F T^{\alpha+\beta}M^{p-1},$$

*respectively. Define the map $\Phi = \Phi_{u_0}$ by*

$$\Phi[u](t) := S_{\mathcal{L}}(t)u_0 + \int_0^t S_{\mathcal{L}}(t - \tau)F(u(\tau))\,d\tau,$$

*for $t \in [0, T]$ and the complete metric space $X = X(T, M)$ by*

$$X := B_{L^r(0,T;L^s)}(M) = \{u \in L^r(0, T; L^s(D)) : \|u\|_{L^r(0,T;L^s)} \leq M\},$$

*equipped with the metric*

$$\mathrm{d}(u, v) := \|u - v\|_{L^r(0,T;L^s)}.$$

*Then $\Phi$ is a contraction mapping from $X$ to itself. Consequently, there exists a unique function $u \in X$ such that $\Phi[u] = u$.*

*Proof.* The proof is based on the standard fixed point argument. Let us take the parameters $r, s$ satisfying

$$r, s \in [1, \infty], \quad \frac{s}{p} \geq 1, \quad \frac{r}{p} \geq 1, \quad 0 < \beta \leq 1 \quad \text{and} \quad \alpha + \beta > 0. \tag{15}$$

Take $M > 0, 0 < T \leq 1$ and $u_0 \in L^\infty(D)$ such that

$$\rho + \delta M \leq M \quad \text{and} \quad \delta < 1. \tag{16}$$

Let $u \in X$. Then

$$\|\Phi[u]\|_{L^r(0,T;L^s)} \leq \|S_{\mathcal{L}}(t)u_0\|_{L^r(0,T;L^s)} + C_F \left\| \int_0^t S_{\mathcal{L}}(t-\tau)(|u(\tau)|^{p-1}u(\tau)) \, d\tau \right\|_{L^r(0,T;L^s)}$$

$$=: I + II.$$

As for the first term, it follows from Hölder's inequality and (1) that

$$I \leq |D|^{\frac{1}{s}} T^{\frac{1}{r}} \|S_{\mathcal{L}}(t)u_0\|_{L^\infty(0,T;L^\infty)} \leq C_{\mathcal{L}} |D|^{\frac{1}{s}} T^{\frac{1}{r}} \|u_0\|_{L^\infty} = \rho.$$

As for the second term, it follows from (1) that

$$II \leq C_{\mathcal{L}} C_F \left\| \int_0^t (t-\tau)^{-\nu(\frac{p}{s} - \frac{1}{s})} \||u(\tau)|^{p-1}u(\tau)\|_{L^{\frac{s}{p}}} \, d\tau \right\|_{L^r((0,T))}$$

$$\leq C_{\mathcal{L}} C_F \left\| \int_0^t (t-\tau)^\alpha \|u(\tau)\|_{L^s}^p \, d\tau \right\|_{L^r((0,T))}$$

$$\leq C_{\mathcal{L}} C_F \|f * g\|_{L^r(\mathbb{R})},$$

where $s \geq 1$ and $s/p \geq 1$ are required and the functions $f, g$ are defined by

$$f(t) = f_T(t) := \begin{cases} t^\alpha & \text{for } 0 < t \leq T, \\ 0 & \text{otherwise,} \end{cases}$$

and

$$g(t) = g_T(t) := \begin{cases} \|u(t)\|_{L^s}^p & \text{for } 0 < t \leq T, \\ 0 & \text{otherwise.} \end{cases}$$

By Young's inequality, we have

$$\|f * g\|_{L^r(\mathbb{R})} \leq \|f\|_{L^{\frac{1}{\beta}}(\mathbb{R})} \|g\|_{L^{\frac{r}{p}}(\mathbb{R})}$$

$$= \left( \alpha \beta^{-1} + 1 \right)^{-\beta} T^{\alpha+\beta} \|u\|_{L^r(0,T;L^s)}^p$$

$$\leq \left( \alpha \beta^{-1} + 1 \right)^{-\beta} T^{\alpha+\beta} M^p,$$

where we require $r/p \geq 1, 0 < \beta \leq 1$ and $\alpha + \beta > 0$ and we calculate

$$\|f\|_{L^{\frac{1}{\beta}}(\mathbb{R})} = \left( \alpha \beta^{-1} + 1 \right)^{-\beta} T^{\alpha+\beta}.$$

Summarizing the estimates obtained above, and then using (14), we have

$$\|\Phi[u]\|_{L^r(0,T;L^s)} \leq \rho + C_{\mathcal{L}} C_F \left( \alpha \beta^{-1} + 1 \right)^{-\beta} T^{\alpha+\beta} M^p = \rho + \delta M \leq M,$$

which implies that $\Phi$ is a mapping from $X$ to itself.

Next, we estimate the metric $d(\Phi[u_1], \Phi[u_2])$. Let $u_1, u_2 \in X$. By using (2), (1) and Hölder's inequality, we have

$$d(\Phi[u_1], \Phi[u_2]) \leq C_{\mathcal{L}} C_F \left\| \int_0^t (t-\tau)^\alpha \| \max_{i=1,2} |u_i(\tau)|^{p-1} |u_1(\tau) - u_2(\tau)| \|_{L^{\frac{s}{p}}} \, d\tau \right\|_{L^r((0,T))}$$

$$\leq 2 C_{\mathcal{L}} C_F \max_{i=1,2} \left\| \int_0^t (t-\tau)^\alpha \|u_i(\tau)\|_{L^s}^{p-1} \|u_1(\tau) - u_2(\tau)\|_{L^s} \, d\tau \right\|_{L^r((0,T))}$$

$$\leq 2 C_{\mathcal{L}} C_F \max_{i=1,2} \|f * h_i\|_{L^r(\mathbb{R})},$$

where $f$ is the same function as above and $h_i$ is defined by

$$h_i(t) = h_{i,T}(t) := \begin{cases} \|u_i(\tau)\|_{L^s}^{p-1}\|u_1(\tau) - u_2(\tau)\|_{L^s} & \text{for } 0 < t \le T, \\ 0 & \text{otherwise.} \end{cases}$$

By Young's inequality and Hölder's inequality, we have

$$\begin{aligned} \mathrm{d}(\Phi[u_1], \Phi[u_2]) &\le 2C_{\mathcal{L}}C_F \max_{i=1,2} \|f * h_i\|_{L^r(\mathbb{R})} \\ &\le 2C_{\mathcal{L}}C_F \max_{i=1,2} \|f\|_{L^{\frac{1}{\beta}}(\mathbb{R})} \|h_i\|_{L^{\frac{r}{p}}(\mathbb{R})} \\ &\le 2C_{\mathcal{L}}C_F \left(\alpha\beta^{-1} + 1\right)^{-\beta} T^{\alpha+\beta} \max_{i=1,2} \|u_i\|_{L^r(0,T;L^s)}^{p-1} \|u_1 - u_2\|_{L^r(0,T;L^s)} \\ &\le 2C_{\mathcal{L}}C_F \left(\alpha\beta^{-1} + 1\right)^{-\beta} T^{\alpha+\beta} M^{p-1} \mathrm{d}(u_1, u_2) = \delta \mathrm{d}(u_1, u_2). \end{aligned}$$

From the second inequality $\delta < 1$ in (16), we show that $\Phi$ is a contraction mapping from $X$ into itself. Therefore, Banach's fixed point theorem allows us to prove that there exists uniquely a function $u \in X$ such that $u = \Phi[u]$. Finally, noting the condition (15) is equivalent to (13) and the condition (16) is also equivalent to (14), we conclude Proposition 3. □

Since $\min\{1/r, \alpha+\beta\}$ is always positive under the condition (13), for any $M > 0$ and $u_0 \in L^\infty(D)$, we can take $T > 0$ sufficiently small so that (14) holds. As a result, we have the following LWP result.

**Corollary 2** (Proposition 1). *Assume that $r, s$ satisfy* (13). *Then, for any $u_0 \in L^\infty(D)$, there exist a time $T = T(u_0) > 0$ and a unique solution $u \in L^r(0, T; L^s(D))$ to* (P'). *Moreover, for any $u_0, v_0 \in L^\infty(D)$, the solutions $u, v$ to* (P') *with $u(0) = u_0$ and $v(0) = v_0$ satisfy the inequality*

$$\|u - v\|_{L^r(0,T';L^s)} \le C\|u_0 - v_0\|_{L^\infty},$$

*where $T' = \min\{T(u_0), T(v_0)\}$.*

*Proof.* The former immediately follows from Proposition 3. The latter is also shown in a similar way to the proof of Proposition 3. In fact, let $u, v \in X$ be solutions to (P') with initial data $u_0$ and $v_0$, respectively. Then

$$\|u - v\|_{L^r(0,T;L^s)} \le C_{\mathcal{L}}|D|^{\frac{1}{s}}T^{\frac{1}{r}}\|u_0 - v_0\|_{L^\infty} + \delta\|u - v\|_{L^r(0,T;L^s)}.$$

If $T$ is sufficiently small so that $\delta < 1$, then we obtain

$$\|u - v\|_{L^r(0,T;L^s)} \le \frac{C_{\mathcal{L}}|D|^{\frac{1}{s}}T^{\frac{1}{r}}}{1-\delta}\|u_0 - v_0\|_{L^\infty}.$$

The proof is finished. □

## C APPLICATIONS TO FNOS AND WNOS

In this appendix, we apply Theorem 1 to FNOs and WNOs for some operator $\mathcal{L}$. Let us recall that, to ensure our quantitative approximation theorem (Theorem 1), the families $\varphi := \{\varphi_n\}_n$ and $\psi := \{\psi_m\}_m$ of functions need to approximate the Green function $G$ of the linear equation $\partial_t u + \mathcal{L}u = 0$ (i.e. the integral kernel $G$ of semigroup generated by $\mathcal{L}$) such that

$$G(t - \tau, x, y) = \sum_{m,n \in \Lambda} c_{m,n}\psi_m(\tau, y)\varphi_n(t, x), \tag{17}$$

$$G_N(t - \tau, x, y) = \sum_{m,n \in \Lambda_N} c_{m,n}\psi_m(\tau, y)\varphi_n(t, x)$$

for $0 \le \tau < t < T$ and $x, y \in D$, where $\Lambda$ is an index set that is either finite or countably infinite and $\Lambda_N$ is a subset of $\Lambda$ with its cardinality $|\Lambda_N| = N \in \mathbb{N}$. The convergence (17) means the sense that

$$E_G(N) := \left\| \|G(t - \tau, x, y) - G_N(t - \tau, x, y)\|_{L_\tau^{r'}(0,T;L_y^{s'})} \right\|_{L_t^r(0,T;L_x^s)} \to 0 \tag{18}$$

and

$$E_G'(N) := \left\| \|G(t,x,y) - G_N(t,x,y)\|_{L_y^{s'}} \right\|_{L_t^r(0,T;L_x^s)} \to 0 \qquad (19)$$

as $N \to \infty$ for some $r, s \in [1, \infty]$ satisfying the condition (3). The following conditions are required for the arguments in this appendix:

$$\left\| \|G(t-\tau,x,y)\|_{L_\tau^{r'}(0,T;L_y^{s'})} \right\|_{L_t^r(0,T;L_x^s)} < \infty, \qquad (20)$$

$$\left\| \|G(t,x,y)\|_{L_y^{s'}} \right\|_{L_t^r(0,T;L_x^s)} < \infty. \qquad (21)$$

**Remark 5.** *Let us give some remarks on* (20) *and* (21).

(a) *It is seen that* (20) *and* (21) *always hold with* $r = s = \infty$ *under Assumption 1 (see Lemma 1 in Appendix D).*

(b) *For example, the operators* $\mathcal{L}$ *appeared in Appendix A.1–A.4 satisfy the Gaussian upper bound* (10)*, and hence, the corresponding Green functions satisfy* (20) *and* (21) *if*

$$\frac{d}{s} + \frac{2}{r} < 2 \quad \text{and} \quad \frac{d}{s} < \frac{2}{r}. \qquad (22)$$

### C.1 FOURIER NEURAL OPERATORS

Before stating FNOs, we provide a remark on partial sums of multiple Fourier series. From the perspective of convergence issues, the rectangular partial sum is preferable to the spherical partial sum. In fact, for $f \in L^1(\mathbb{T}^d)$, the Fourier coefficient of $f$ is defined by

$$c_n = \hat{f}(n) := \int_{\mathbb{T}^d} f(x) e^{-2\pi i n x} \, dx, \quad n = (n_1, n_2, \cdots, n_d) \in \mathbb{Z}^d,$$

where $\mathbb{T}^d = \mathbb{R}^d/\mathbb{Z}^d$ is the $d$-dimensional torus, $x = (x_1, x_2, \ldots, x_d)$ and $nx := n_1 x_1 + \cdots + n_d x_d$. We define the rectangular partial sum of multiple Fourier series of $f$ by

$$S_N(f)(x) := \sum_{|n_1|, \cdots, |n_d| \leq N} c_n e^{2\pi i n x}, \quad N \in \mathbb{N}.$$

Then we have the following:

**Theorem 2** ((Sjölin, 1971; Fefferman, 1971)). *Let* $q > 1$. *Then*

$$\lim_{N \to \infty} S_N(f)(x) = f(x) \quad \text{a.e. } x \quad \text{and} \quad \lim_{N \to \infty} S_N(f) = f \quad \text{in } L^q(\mathbb{T}^d),$$

*for any* $f \in L^q(\mathbb{T}^d)$.

This theorem can be easily extended to periodic functions $f$ on a general $d$-dimensional rectangle with usual modifications. In contrast, the spherical partial sum of multiple Fourier series of $f \in L^q(\mathbb{T}^d)$ converges to $f$ in $L^q$ only if $q = 2$, and for its pointwise convergence, it remains an open problem even when $q = 2$.

Next, we mention the approximation theorem on FNOs. Here, we apply the Fourier expansion with respect to both time $t$ and space $x$ to define FNOs. This type of FNO is used in the implementation to approximate some PDEs (see e.g. Kovachki et al. (2023, Section 7.3)).

For simplicity, we consider only the case $d = 1$, $r = s = 2$ and $\mathcal{L}$ is one of Appendixes A.1–A.4 (which implies $\nu = d/2$). Then the condition (22) is satisfied and Theorem 1 can be applied to FNOs under the condition (3), i.e., $1 < p < 1 + \frac{4}{d+2}$, if we can show that (18) and (19) hold for the Fourier basis.

In fact, we consider the nonlinear parabolic PDEs (P), where $D$ is a bounded interval in $\mathbb{R}$. As $G$ is not a periodic function, a zero extension of $G$ is considered here, so that the multiple Fourier series can be applied. We take $\tilde{D}$ as a bounded interval which includes $D$ and with its length $L$, and we define the zero extension $\tilde{G}$ of $G = \tilde{G}(t-\tau,x,y)$ to $[-T, 2T] \times [-T, 2T] \times \tilde{D} \times \tilde{D}$ by

$$\tilde{G}(t-\tau,x,y) = \begin{cases} G(t-\tau,x,y) & \text{if } 0 \leq \tau < t < T, \ x,y \in D \\ 0 & \text{otherwise.} \end{cases}$$

Then, thanks to (20) and (21), $\tilde{G}$ has the multiple Fourier series

$$\tilde{G}(t-\tau,x,y) = \lim_{N\to\infty} \sum_{m,n\in\Lambda_N} c_{m,n} e^{\frac{2\pi in_1 t}{3T}} e^{\frac{2\pi im_1 \tau}{3T}} e^{\frac{2\pi in_2 x}{L}} e^{\frac{2\pi im_2}{y}}$$

for almost everywhere in $(t,\tau,x,y) \in [-T,2T] \times [-T,2T] \times \tilde{D} \times \tilde{D}$ and it also converges in $L^2([-T,2T] \times [-T,2T] \times \tilde{D} \times \tilde{D})$, where $\Lambda_N := \{(m,n) = (m_1,m_2,n_1,n_2) \in \mathbb{Z}^4 : |m_1|,|m_2|,|n_1|,|n_2| \le N^{1/4}\}$ and $c_n$ is the Fourier coefficient given by

$$c_{m,n} := \int_{-T}^{2T} \int_{-T}^{2T} \int_{\tilde{D}} \int_{\tilde{D}} \tilde{G}(t-\tau,x,y) e^{-\frac{2\pi in_1 t}{3T}} e^{-\frac{2\pi in_2 \tau}{3T}} e^{-\frac{2\pi in_3 x}{L}} e^{-\frac{2\pi in_4 y}{L}} \, dt d\tau dx dy.$$

Hence, putting $\varphi_n(t,x) := e^{\frac{2\pi in_1 t}{3T}} e^{\frac{2\pi in_2 x}{L}}$ and $\psi_m(\tau,y) := e^{\frac{2\pi im_1 \tau}{3T}} e^{\frac{2\pi im_2 y}{L}}$, we have the expansion (17) of $G$ with (18) and (19). Therefore, by choosing these $\varphi = \{\varphi_n\}_n$ and $\psi = \{\psi_m\}_m$ in Definition 1, we have the following result on the FNO as a corollary of Theorem 1.

**Corollary 3.** *Let $d = 1$, $r = s = 2$, $\mathcal{L}$ be one of Appendixes A.1–A.4 ($\nu = 1/2$), $\varphi = \{\varphi_n\}_n$ and $\psi = \{\psi_m\}_m$ as above, and $F$ satisfy Assumption 2 with $1 < p < 7/3$. Then for any $R > 0$, there exists a time $T > 0$ such that the following statement holds: For any $\epsilon \in (0,1)$, there exist a depth $L$, the number of neurons $H$, a rank $N$ and an FNO $\Gamma \in \mathcal{NO}_{N,\varphi,\psi}^{L,H,ReLU}$ such that*

$$\sup_{u_0 \in B_{L^\infty}(R)} \|\Gamma^+(u_0) - \Gamma(u_0)\|_{L^2(0,T;L^2)} \le \epsilon.$$

*Moreover, $L = L(\Gamma)$ and $H = H(\Gamma)$ satisfy*

$$L(\Gamma) \le C(\log(\epsilon^{-1}))^2 \quad and \quad H(\Gamma) \le C\epsilon^{-1}(\log(\epsilon^{-1}))^2,$$

*where $C > 0$ is a constant depending on $\nu, M, F, D, p, d, r, s, R$ and $\mathcal{L}$.*

### C.2 HAAR WAVELET NEURAL OPERATORS

A wavelet is an oscillatory function with zero mean, used to analyze signals or functions locally at different scales. There are various types of wavelets, such as Haar, Meyer, Morlet, and Shannon and the systems constructed by these wavelets are unconditional bases for $L^q(\mathbb{R}^d)$ with $1 < q < \infty$ (see e.g. Meyer (1992); Wojtaszczyk (1997)). Wavelets can also be defined on domains other than on $\mathbb{R}^d$ (see e.g. Triebel (2008)). Several results on wavelets for mixed Lebesgue spaces are also known (see e.g. Georgiadis et al. (2017); Pandey & Viswanathan (2024); Torres & Ward (2015)).

In this appendix, we use the fact that the multiparameter Haar system is an unconditional basis for the mixed Lebesgue spaces $L^{\vec{q}}((0,1)^d)$ with $\vec{q} = (q_1,\ldots,q_d) \in (1,\infty)^d$ by Pandey & Viswanathan (2024, Theorem 4.9).

The set of dyadic intervals in $[0,1]$ is defined by

$$\mathcal{D} := \left\{ I_{j,k} := \left[ \frac{k}{2^{j-1}}, \frac{k+1}{2^{j-1}} \right) : 0 \le k < 2^{j-1}, \, j \in \mathbb{N} \right\}.$$

For $I \in \mathcal{D}$, we denote by $I_{\text{left}}$ and $I_{\text{right}}$ the left and right halves of $I$, respectively, and we define the Haar function $h_I$ by $h_I := \chi_{I_{\text{left}}} - \chi_{I_{\text{right}}}$. Here, $\chi_I$ is the characteristic function of an interval $I$. The set of dyadic hyper-rectangles in $[0,1]^d$ is defined by

$$\mathcal{R}_d := \{\mathcal{I} := I_1 \times I_2 \times \cdots \times I_d : I_1, I_2, \cdots, I_d \in \mathcal{D}\}.$$

For $\mathcal{I} \in \mathcal{R}_d$, we define

$$h_{\mathcal{I}}(x) := h_{I_1}(x_1) h_{I_2}(x_2) \cdots h_{I_d}(x_d)$$

for $x = (x_1, x_2, \ldots, x_d) \in [0,1]^d$. Then the multiparameter Haar system defined by $\{h_{\mathcal{I}} : \mathcal{I} \in \mathcal{R}_d\}$ is an unconditional basis for the mixed Lebesgue spaces $L^{\vec{q}}((0,1)^d)$ with $\vec{q} = (q_1,\ldots,q_d) \in (1,\infty)^d$ (see Pandey & Viswanathan (2024, Theorem 4.8)).

Let $\tilde{G}$ be the zero extension of the Green function $G$ to a function on $[0,1]^{2d+2}$ ($= [0,1]^{1+d+1+d}$) with respect to $t, x, \tau, y$. Then we can apply Pandey & Viswanathan (2024, Theorem 4.8) to $\tilde{G}$ and then

$$\tilde{G}(t-\tau,x,y) = \lim_{N\to\infty} \sum_{\mathcal{I}\in\mathcal{R}_{2d+2,N}} c_{\mathcal{I}} h_{\mathcal{I}_t}(t) h_{\mathcal{I}_x}(x) h_{\mathcal{I}_\tau}(\tau) h_{\mathcal{I}_y}(y)$$

in the sense of mixed Lebesgue norms of $L^{\vec{q}}((0,1)^{2d+2})$ with all $\vec{q} = (q_1, \ldots, q_d) \in (1, \infty)^{2d+2}$, where $\mathcal{R}_{2d+2,N}$ is a subset of $\mathcal{R}_{2d+2}$ with its cardinality $|\mathcal{R}_{2d+2,N}| = N$, $c_{\mathcal{I}}$ are wavelet coefficients, and $\mathcal{I} = \mathcal{I}_t \times \mathcal{I}_x \times \mathcal{I}_\tau \times \mathcal{I}_y \in \mathcal{D} \times \mathcal{D}^d \times \mathcal{D} \times \mathcal{D}^d$. Hence, putting $c_{m,n} = c_{\mathcal{I}}$, $\varphi_n(t, x) = h_{\mathcal{I}_t}(t) h_{\mathcal{I}_x}(x)$ and $\psi_m(\tau, y) = h_{\mathcal{I}_\tau}(\tau) h_{\mathcal{I}_y}(y)$ with $m = (k_1, j_1) \in (\mathbb{N} \cup \{0\})^{d+1} \times \mathbb{N}^d$ and $n = (k_2, j_2) \in (\mathbb{N} \cup \{0\})^{d+1} \times \mathbb{N}^d$, we have the expansion (17) of $G$ with (18) and (19), where $\Lambda_N$ is an index set of $(k_1, j_1, k_2, j_2)$ corresponding to $\mathcal{R}_{2d+2,N}$ with its cardinality $|\Lambda_N| = N$. Therefore, by choosing these $\varphi = \{\varphi_n\}_n$ and $\psi = \{\psi_m\}_m$ in Definition 1, we have the following result on the Haar WNO as a corollary of Theorem 1.

**Corollary 4.** *Let $d \in \mathbb{N}$. Suppose $\mathcal{L}$ is one of Appendixes A.1–A.4 ($\nu = d/2$) and $F$ satisfies Assumption 2. Let $r, s \in (1, \infty)$ satisfy the conditions (3) and (22), and let $\varphi = \{\varphi_n\}_n$ and $\psi = \{\psi_m\}_m$ be as above. Then, for any $R > 0$, there exists a time $T > 0$ such that the following statement holds: For any $\epsilon \in (0, 1)$, there exist a depth $L$, the number of neurons $H$, a rank $N$ and a Haar WNO $\Gamma \in \mathcal{NO}_{N,\varphi,\psi}^{L,H,ReLU}$ such that*

$$\sup_{u_0 \in B_{L^\infty}(R)} \|\Gamma^+(u_0) - \Gamma(u_0)\|_{L^r(0,T;L^s)} \le \epsilon.$$

*Moreover, $L = L(\Gamma)$ and $H = H(\Gamma)$ satisfy*

$$L(\Gamma) \le C(\log(\epsilon^{-1}))^2 \quad and \quad H(\Gamma) \le C\epsilon^{-1}(\log(\epsilon^{-1}))^2,$$

*where $C > 0$ is a constant depending on $\nu, M, F, D, p, d, r, s, R$ and $\mathcal{L}$.*

Note that the previous papers of WNOs such as Gupta et al. (2021); Tripura & Chakraborty (2023) have not employed Haar wavelet in the context of learning solution operators for PDEs. Here, we just chosen the Haar wavelet as a concrete example of basis expansion of the Green function, and the discussions choosing other wavelets could be possible.

## D  PROOF OF THEOREM 1

Suppose that $\mathcal{L}$ and $F$ satisfy Assumptions 1 and 2, respectively, and the parameters $r, s$ satisfy the condition (3). The conditions for the parameters $R, T, M, M'$ that appear in the proof are specified here. Let us take $R, M, M' > 0$ and $T \in (0, 1]$ such that

$$\begin{cases} C_{\mathcal{L}}|D|^{\frac{1}{s}} T^{\frac{1}{r}} R + (\alpha\beta^{-1} + 1)^{-\beta} C_{\mathcal{L}} C_F T^{\alpha+\beta} M^p \le M, \\ 2C_{\mathcal{L}} R + 2C_{\mathcal{L}} T(1 + C_F M'^p) \le M', \\ T^{\frac{1}{r}} |D|^{\frac{1}{s}} M' \le M, \end{cases} \tag{23}$$

where $\alpha$ and $\beta$ are the constants defined in (12). The first condition in (23) ensures the LWP for (P') (Propositions 1 and 2). See Proposition 3 and (14) in Appendix B for more details. The second condition ensures that the approximate mappings $\Phi_{N_k}$ and $\Phi_{N_k,net}$ in the proof map from $B_{L^\infty(0,T;L^\infty)}(M')$ into itself. The third condition ensures $B_{L^\infty(0,T;L^\infty)}(M') \subset B_{L^r(0,T;L^s)}(M)$. We note to take the parameters in the following order: For any $R, M > 0$, we can take a sufficiently large $M'$ and a sufficiently small $T$ so that (23) holds.

To begin with, we prepare the following lemma.

**Lemma 1.** *Under Assumption 1, we have*

$$\left\| \|G(t - \tau, x, y)\|_{L^1_\tau(0,T;L^1_y)} \right\|_{L^\infty_t(0,T;L^\infty_x)} \le C_{\mathcal{L}} T,$$

$$\left\| \|G(t, x, y)\|_{L^1_y} \right\|_{L^\infty_t(0,T;L^\infty_x)} \le C_{\mathcal{L}},$$

*for any $T \in (0, 1]$.*

*Proof.* From Assumption 1 and Iwabuchi et al. (2021, Lemma B.1), we see that

$$\|G(t - \tau, x, y)\|_{L^\infty_x L^1_y} = \|S_{\mathcal{L}}(t - \tau)\|_{L^\infty \to L^\infty} \le C_{\mathcal{L}},$$

which implies that

$$\left\| \|G(t-\tau,x,y)\|_{L^1_\tau(0,T;L^1_y)} \right\|_{L^\infty_t(0,T;L^\infty_x)} \le C_\mathcal{L} T.$$

Similarly, as

$$\|G(t,x,y)\|_{L^\infty_x L^1_y} = \|S_\mathcal{L}(t)\|_{L^\infty \to L^\infty} \le C_\mathcal{L},$$

we also have

$$\left\| \|G(t,x,y)\|_{L^1_y(D)} \right\|_{L^\infty_t(0,T;L^\infty_x)} \le C_\mathcal{L}.$$

The proof of Lemma 1 is finished. $\qquad\square$

We divide the proof of Theorem 1 into four steps.

**Step 1.** We define $\Phi_N$ by

$$\Phi_N[u](t,x) := \int_D G_N(t,x,y)u_0(y)dy + \int_0^t \int_D G_N(t-\tau,x,y)F(u(\tau,y))d\tau dy.$$

Since $E_G(N), E'_G(N) \to 0$ as $N \to \infty$ with $r', s' \ge 1$ and $D \subset \mathbb{R}^d$ is bounded, we see that

$$\left\| \|G(t-\tau,x,y) - G_N(t-\tau,x,y)\|_{L^1_\tau(0,T;L^1_y)} \right\|_{L^r_t(0,T;L^s_x)} \to 0,$$

$$\left\| \|G(t,x,y) - G_N(t,x,y)\|_{L^1_y} \right\|_{L^r_t(0,T;L^s_x)} \to 0,$$

as $N \to \infty$, and hence, there exists a subsequence $\{N_k\}_{k\in\mathbb{N}} \subset \mathbb{N}$ such that

$$\|G_{N_k}(t-\tau,x,y)\|_{L^1_\tau(0,T;L^1_y)} \to \|G(t-\tau,x,y)\|_{L^1_\tau(0,T;L^1_y)},$$

$$\|G_{N_k}(t,x,y)\|_{L^1_y} \to \|G(t,x,y)\|_{L^1_y},$$

for almost everywhere $t \in (0,T)$ and $x \in D$ as $k \to \infty$. Then, for any $\epsilon > 0$, there exists $k_\epsilon \in \mathbb{N}$ such that for any $k \ge k_\epsilon$,

$$\left\| \|G_{N_k}(t-\tau,x,y)\|_{L^1_\tau(0,T;L^1_y)} \right\|_{L^\infty_t(0,T;L^\infty_x)} \le 2C_\mathcal{L} T,$$

$$\left\| \|G_{N_k}(t,x,y)\|_{L^1_y} \right\|_{L^\infty_t(0,T;L^\infty_x)} \le 2C_\mathcal{L},$$

$$E_G(N_k), E'_G(N_k) \le \epsilon,$$

where the first and second inequalities follow from Lemma 1 and the last one follows from Assumption 3.

**Lemma 2.** *For any $u_0 \in B_{L^\infty}(R)$ and $k \ge k_\epsilon$, the following statements hold:*

(i) $\Phi_{N_k}$ *is a map from $B_{L^\infty(0,T;L^\infty)}(M')$ to itself.*

(ii) *There exists a constant $C_1 > 0$ such that*

$$\|\Phi[u] - \Phi_{N_k}[u]\|_{L^r(0,T;L^s)} \le C_1\epsilon,$$

*for $u \in B_{L^\infty(0,T;L^\infty)}(M')$. Here, $C_1 = |D|^{\frac{1}{s}}R + C_F M^p$.*

*Proof.* Let $u_0 \in B_{L^\infty}(R)$ and $k \ge k_\epsilon$. We first prove (i). By Hölder's inequality and the second condition in (23), for any $u \in B_{L^\infty(0,T;L^\infty)}(M')$, we estimate

$$\|\Phi_{N_k}[u]\|_{L^\infty_t(0,T;L^\infty_x)} \le \left\| \|G_{N_k}(t,x,y)\|_{L^1_y} \right\|_{L^\infty_t(0,T;L^\infty_x)} \|u_0\|_{L^\infty}$$

$$+ \left\| \|G_{N_k}(t-\tau,x,y)\|_{L^1_\tau(0,T;L^1_y)} \right\|_{L^\infty_t(0,T;L^\infty_x)} \|F(u)\|_{L^\infty(0,T;L^\infty)}$$

$$\le 2C_\mathcal{L} R + 2C_\mathcal{L} T C_F M'^p \le M'.$$

Next, we prove (ii). By Hölder's inequality and the third condition in (23), for any $u \in B_{L^\infty(0,T;L^\infty)}(M')$, we have

$$\left| \Phi[u](t,x) - \Phi_{N_k}[u](t,x) \right| \leq \|G(t,x,y) - G_{N_k}(t,x,y)\|_{L_y^{s'}} \|u_0\|_{L^s}$$
$$+ \|G(t-\tau,x,y) - G_{N_k}(t-\tau,x,y)\|_{L_\tau^{r'}(0,T;L_y^{s'})} \|F(u)\|_{L^r(0,T;L^s)},$$

which implies that

$$\|\Phi[u] - \Phi_{N_k}[u]\|_{L^r(0,T;L^s)}$$
$$\leq \left\| \|G(t,x,y) - G_N(t,x,y)\|_{L_y^{s'}} \right\|_{L_t^r(0,T;L_x^s)} \|u_0\|_{L^s}$$
$$+ \left\| \|G(t-\tau,x,y) - G_N(t-\tau,x,y)\|_{L_\tau^{r'}(0,T;L_y^{s'})} \right\|_{L_t^r(0,T;L_x^s)} \|F(u)\|_{L^r(0,T;L^s)}$$
$$\leq E_G'(N_k)\|u_0\|_{L^s} + E_G(N_k)\|F(u)\|_{L^r(0,T;L^s)}$$
$$\leq \epsilon |D|^{\frac{1}{s}} \|u_0\|_{L^\infty} + \epsilon \|F(u)\|_{L^r(0,T;L^s)}$$
$$\leq \left( |D|^{\frac{1}{s}} R + C_F M^p \right) \epsilon$$
$$= C_1 \epsilon,$$

where we note that $B_{L^\infty(0,T;L^\infty)}(M') \subset B_{L^r(0,T;L^s)}(M)$. Therefore, (ii) is proved. Thus, the proof of Lemma 2 is finished. □

**Step 2.** We define the map $\Phi_{N_k,net}$ by

$$\Phi_{N_k,net}[u](t,x) := \int_D G_{N_k}(t,x,y)u_0(y)dy + \int_0^t \int_D G_{N_k}(t-\tau,x,y)F_{net}(u(\tau,y))d\tau dy, \quad (24)$$

where $F_{net} : \mathbb{R} \to \mathbb{R}$ is a ReLU neural network. Here, as $F|_{(-M',M')} \in W^{1,\infty}(-M',M')$ from Assumption 2, we can apply the approximation result by Gühring et al. (2020, Theorem 4.1) to see that there exists a ReLU neural network $F_{net} : \mathbb{R} \to \mathbb{R}$ such that

$$\|F - F_{net}\|_{L^\infty(-M,M)} \leq \epsilon, \quad (25)$$

where the depth $L = L(F_{net})$ and the number $H = H(F_{net})$ of neurons are estimated as

$$\begin{cases} L(F_{net}) \leq C \log_2 \left( \epsilon^{-1} \right), \\ H(F_{net}) \leq C \epsilon^{-1} \log_2 \left( \epsilon^{-1} \right), \end{cases} \quad (26)$$

where $C = C(M', F) > 0$ is a constant depending on $M', F$.

**Lemma 3.** *For any $u_0 \in B_{L^\infty}(R)$ and $k \geq k_\epsilon$, the following statements hold:*

(i) *$\Phi_{N_k,net}$ is a map from $B_{L^\infty(0,T;L^\infty)}(M')$ to itself.*

(ii) *There exists a constant $C_2 > 0$ such that*

$$\|\Phi_{N_k}[u] - \Phi_{N_k,net}[u]\|_{L^r(0,T;L^s)} \leq C_2 \epsilon.$$

*for $u \in B_{L^\infty(0,T;L^\infty)}(M')$. Here, $C_2 = 2C_{\mathcal{L}}|D|^{\frac{1}{s}} T^{1+\frac{1}{r}}$.*

*Proof.* The proof follows the same lines with Lemma 2. Let $u_0 \in B_{L^\infty}(R)$ and $k \geq k_\epsilon$. By Hölder's inequality and Assumption 2, for any $u \in B_{L^\infty(0,T;L^\infty)}(M')$, we estimate

$$\|\Phi_{N_k,net}[u]\|_{L^\infty(0,T;L^\infty)} \leq \left\| \|G_{N_k}(t,x,y)\|_{L_y^1} \right\|_{L_t^\infty(0,T;L_x^\infty)} \|u_0\|_{L^\infty}$$
$$+ \left\| \|G_{N_k}(t-\tau,x,y)\|_{L_\tau^1(0,T;L_y^1)} \right\|_{L_t^\infty(0,T;L_x^\infty)} \|F_{net}(u)\|_{L^\infty(0,T;L^\infty)}$$
$$\leq 2C_{\mathcal{L}} R + 2C_{\mathcal{L}} T(1 + C_F M'^p) \leq M',$$

where we used the inequality

$$\|F_{net}(u)\|_{L^\infty(0,T;L^\infty)} \le \|F(u) - F_{net}(u)\|_{L^\infty(0,T;L^\infty)} + \|F(u)\|_{L^\infty(0,T;L^\infty)}$$
$$\le \epsilon + C_F M'^p \le 1 + C_F M'^p.$$

Hence, (i) is proved.

Next, we prove (ii). By Hölder's inequality, for any $u \in B_{L^\infty(0,T;L^\infty)}(M')$, we estimate

$$\left| \Phi_{N_k}[u](t,x) - \Phi_{N_k,net}[u](t,x) \right|$$
$$\le \|G_{N_k}(t-\tau,x,y)\|_{L^1_\tau(0,T;L^1_y)} \|F(u) - F_{net}(u)\|_{L^\infty(0,T;L^\infty)}$$
$$\le 2C_\mathcal{L}T \|F(u) - F_{net}(u)\|_{L^\infty(0,T;L^\infty)} \le 2C_\mathcal{L}T\epsilon.$$

Hence, we obtain

$$\|\Phi_{N_k}[u] - \Phi_{N_k,net}[u]\|_{L^r(0,T;L^s)} \le 2C_\mathcal{L}|D|^{\frac{1}{s}}T^{1+\frac{1}{r}}\epsilon = C_2\epsilon.$$

Therefore, (ii) is proved. Thus, the proof of Lemma 3 is finished. $\square$

**Step 3.** We define $\Gamma : B_{L^\infty}(R) \to B_{L^r(0,T;L^s)}(M)$ by

$$\Gamma(u_0) := \Phi^{[J]}_{N_k,net}[0] \quad \text{for } u_0 \in B_{L^\infty}(R). \tag{27}$$

**Lemma 4.** *Let* $J = \lceil \frac{\log(1/\epsilon)}{\log(1/\delta)} \rceil \in \mathbb{N}$. *Then there exists a constant* $C_3 > 0$ *such that for any* $u_0 \in B_{L^\infty}(R)$
$$\|\Gamma^+(u_0) - \Gamma(u_0)\|_{L^r(0,T;L^s)} \le C_3\epsilon,$$
*where* $C_3 = M + \frac{C_1 + C_2}{1-\delta}$.

*Proof.* By the triangle inequality, we have

$$\|\Gamma^+(u_0) - \Gamma(u_0)\|_{L^r(0,T;L^s)} \le \|\Gamma^+(u_0) - \Phi^{[J]}[0]\|_{L^r(0,T;L^s)} + \|\Phi^{[J]}[0] - \Gamma(u_0)\|_{L^r(0,T;L^s)}.$$

As to the first term, since $\Phi : B_{L^r(0,T;L^s)}(M) \to B_{L^r(0,T;L^s)}(M)$ is $\delta$-contractive and $u = \Gamma^+(u_0)$ is the fixed point of $\Phi$ from Proposition 2, we have

$$\|\Gamma^+(u_0) - \Phi^{[J]}[0]\|_{L^r(0,T;L^s)} = \|\Phi^{[J]}[u] - \Phi^{[J]}[0]\|_{L^r(0,T;L^s)}$$
$$\le \delta^J \|u - 0\|_{L^r(0,T;L^s)} \le M\delta^J \le M\epsilon. \tag{28}$$

As to the second term, we see that

$$\|\Phi^{[J]}[0] - \Gamma(u_0)\|_{L^r(0,T;L^s)} = \|\Phi^{[J]}[0] - \Phi^{[J]}_{N_k,net}[0]\|_{L^r(0,T;L^s)}$$

$$\le \sum_{j=1}^{J} \left\| \left( \Phi^{[J-j+1]} \circ \Phi^{[j-1]}_{N_k,net} \right)[0] - \left( \Phi^{[J-j]} \circ \Phi^{[j]}_{N_k,net} \right)[0] \right\|_{L^r(0,T;L^s)} \tag{29}$$

$$\le \sum_{j=1}^{J} \delta^{J-j} \left\| \left( \Phi \circ \Phi^{[j-1]}_{N_k,net} \right)[0] - \Phi^{[j]}_{N_k,net}[0] \right\|_{L^r(0,T;L^s)} \tag{30}$$

$$= \sum_{j=1}^{J} \delta^{J-j} \|\Phi[u_{j-1,N}] - \Phi_{N_k,net}[u_{j-1,N}]\|_{L^r(0,T;L^s)}, \tag{31}$$

where $u_{j,N} := \Phi^{[j]}_{N_k,net}[0]$ and $u_{j-1,N} \in B_{L^\infty(0,T;L^\infty)}(M') \subset B_{L^r(0,T;L^s)}(M)$ by the third condition in (23). By Lemma 2 and Lemma 3, we see that

$$\|\Phi[u_{j,N}] - \Phi_{N_k,net}[u_{j,N}]\|_{L^r(0,T;L^s)},$$
$$\le \|\Phi[u_{j-1,N}] - \Phi_{N_k}[u_{j-1,N}]\|_{L^r(0,T;L^s)} + \|\Phi_{N_k}[u_{j-1,N}] - \Phi_{N_k,net}[u_{j-1,N}]\|_{L^r(0,T;L^s)}$$
$$\le (C_1 + C_2)\epsilon.$$

Hence,

$$\|\Phi^{[J]}[0] - \Gamma(u_0)\|_{L^r(0,T;L^s)} \leq \sum_{j=1}^{J} \delta^{J-j}(C_1 + C_2)\epsilon \leq \sum_{j=0}^{\infty} \delta^j(C_1 + C_2)\epsilon = \frac{C_1 + C_2}{1 - \delta}\epsilon. \quad (32)$$

Therefore, by (28) and (32), we conclude that

$$\|\Gamma^+(u_0) - \Gamma(u_0)\|_{L^r(0,T;L^s)} \leq \|\Gamma^+(u_0) - \Phi^{[J]}[0]\|_{L^r(0,T;L^s)} + \|\Phi^{[J]}[0] - \Gamma(u_0)\|_{L^r(0,T;L^s)}$$

$$\leq \left\{ M + \frac{C_1 + C_2}{1 - \delta} \right\} \epsilon = C_3\epsilon.$$

Thus, the proof of Lemma 4 is finished. □

**Final step.** Finally, it is sufficient to represent the approximate operator $\Gamma$ given in (27) as a neural operator in the form of Definition 1 and to provide its quantitative estimates.

**Lemma 5.**

$$\Gamma \in \mathcal{NO}_{N,\varphi,\psi}^{L,H,ReLU},$$

where $L = L(\Gamma)$, $H = H(\Gamma)$ satisfies

$$L(\Gamma) \leq C(\log(\epsilon^{-1}))^2 \quad and \quad H(\Gamma) \leq C\epsilon^{-1}(\log(\epsilon^{-1}))^2,$$

where $C > 0$ is a constant depending on $\nu, M, F, D, p, d, r, s, R, \mathcal{L}$, and $N = N(\Gamma)$ satisfies

$$E_G(N) \leq \epsilon \quad and \quad E_G'(N) \leq \epsilon.$$

*Proof.* Let $N \geq N_k$. Then we have

$$E_G(N) \leq E_G(N_k) \leq \epsilon,$$
$$E_G'(N) \leq E_G'(N_k) \leq \epsilon.$$

We see that

$$\Phi_{N_k,net}[u](t,x)$$

$$= \int_D G_{N_k}(t,x,y)u_0(y)dy + \int_0^t \int_D G_{N_k}(t-\tau,x,y)F_{net}(u(\tau,y))d\tau dy$$

$$= \sum_{n,m\in\Lambda_{N_k}} c_{n,m}\langle\psi_m(0,\cdot),u_0\rangle\varphi_n(t,x) + \sum_{n,m\in\Lambda_{N_k}} c_{n,m}\langle\psi_m,F_{net}(u)\rangle\varphi_n(t,x)$$

$$= \sum_{n,m\in\Lambda_N} \tilde{c}_{n,m}\langle\psi_m(0,\cdot),u_0\rangle\varphi_n(t,x) + \sum_{n,m\in\Lambda_N} \tilde{c}_{n,m}\langle\psi_m,F_{net}(u)\rangle\varphi_n(t,x),$$

where

$$\tilde{c}_{n,m} := \begin{cases} c_{n,m} & \text{if } n,m \in \Lambda_{N_k}, \\ 0 & \text{otherwise.} \end{cases}$$

We denote by

$$\tilde{u}_1(t,x) := \sum_{n,m\in\Lambda_N} \tilde{c}_{n,m}\langle\psi_m(0,\cdot),u_0\rangle\varphi_n(t,x).$$

Then we see that

$$\Gamma(u_0)(t,x) := \Phi_{N_k,net}^{[J]}[0](t,x) = v_J(t,x),$$

where $v_0 := 0$ and

$$v_{j+1}(t,x) := \tilde{u}_1(t,x) + \sum_{n,m\in\Lambda_N} \tilde{c}_{n,m}\langle\psi_m,F_{net}(v_j)\rangle\varphi_n(t,x), \quad j = 0, \cdots, J-1.$$

We define $\tilde{K}_N^{(0)} : L^\infty(D) \to L^r(0,T;L^s(D))^2$ by

$$(\tilde{K}_N^{(0)}u_0)(t,x) := \sum_{n,m\in\Lambda_N} \tilde{C}_{n,m}^{(0)}\langle\psi_m(0,\cdot),u_0\rangle\varphi_n(t,x) = \begin{pmatrix} \tilde{u}_1(t,x) \\ \tilde{u}_1(t,x) \end{pmatrix},$$

where $\tilde{C}_{n,m}^{(0)} = \begin{pmatrix} \tilde{c}_{n,m} \\ \tilde{c}_{n,m} \end{pmatrix} \in \mathbb{R}^{2 \times 1}$. We also define $\tilde{b}_N^{(0)} \in L^r(0, T; L^s(D))^2$ by

$$\tilde{b}_N^{(0)}(t, x) := \begin{pmatrix} \displaystyle\sum_{n,m \in \Lambda_N} \tilde{c}_{n,m}^{(0)} \langle \psi_m, F_{net}(0) \rangle \varphi_n(t, x) \\ 0 \end{pmatrix} = \sum_{n \in \Lambda_N} \tilde{b}_{N,n}^{(0)} \varphi_n(t, x),$$

where

$$\tilde{b}_{N,n}^{(0)} = \begin{pmatrix} \tilde{c}_{n,m}^{(0)} \langle \psi_m, F_{net}(0) \rangle \\ 0 \end{pmatrix} \in \mathbb{R}^{2 \times 1}.$$

Then, we see that

$$(\tilde{K}_N^{(0)} u_0)(t, x) + \tilde{b}_N^{(0)}(t, x) = \begin{pmatrix} v_1(t, x) \\ \tilde{u}_1(t, x) \end{pmatrix},$$

which corresponds to the input layer.

Next, we define

$$\tilde{W} = \begin{pmatrix} 0 & 1 \\ 0 & 1 \end{pmatrix} \in \mathbb{R}^{2 \times 2}$$

and

$$(\tilde{K}_N u)(t, x) := \sum_{n,m \in \Lambda_N} \tilde{C}_{n,m} \langle \psi_m, u \rangle \varphi_n(t, x) = \begin{pmatrix} \displaystyle\sum_{n,m \in \Lambda_N} \tilde{c}_{n,m} \langle \psi_m, u_1 \rangle \varphi_n(t, x) \\ 0 \end{pmatrix}$$

for $u = (u_1, u_2) \in L^r(0, T; L^s(D))^2$, where $\tilde{C}_{n,m} = \begin{pmatrix} \tilde{c}_{n,m} & 0 \\ 0 & 0 \end{pmatrix} \in \mathbb{R}^{2 \times 2}$. We also define $\tilde{F}_{net} : \mathbb{R}^2 \to \mathbb{R}^2$ by

$$\tilde{F}_{net}(u) = \begin{pmatrix} F_{net}(u_1) \\ u_2 \end{pmatrix}, \quad u = (u_1, u_2) \in \mathbb{R}^2,$$

which is a ReLU neural network with (26) (Note that ReLU neural networks represent the identity map). Then we write

$$\left[ (\tilde{W} + \tilde{K}_N) \circ \tilde{F}_{net} \begin{pmatrix} v_j \\ \tilde{u}_1 \end{pmatrix} \right](t, x) = \tilde{W} \begin{pmatrix} F_{net}(v_j)(t, x) \\ \tilde{u}_1(t, x) \end{pmatrix} + \tilde{K}_N \begin{pmatrix} F_{net}(v_j) \\ \tilde{u}_1 \end{pmatrix}(t, x)$$

$$= \begin{pmatrix} \tilde{u}_1(t, x) + \displaystyle\sum_{n,m \in \Lambda_N} \tilde{c}_{n,m} \langle \psi_m, F_{net}(v_j) \rangle \varphi_n(t, x) \\ \tilde{u}_1(t, x) \end{pmatrix}$$

$$= \begin{pmatrix} v_{j+1}(t, x) \\ \tilde{u}_1(t, x) \end{pmatrix} \quad \text{for } j = 1, \ldots, J - 1,$$

which corresponds to the hidden layer.

Finally, denoting by

$$\tilde{W}' := (1, 0) \in \mathbb{R}^{1 \times 2},$$

which corresponds to the last layer, we obtain

$$\Gamma(u_0) = \tilde{W}' \circ \left[ \underbrace{(\tilde{W} + \tilde{K}_N) \circ \tilde{F}_{net} \circ \cdots \circ (\tilde{W} + \tilde{K}_N) \circ \tilde{F}_{net}}_{J-1} \right] \circ (\tilde{K}_N^{(0)} + \tilde{b}_N^{(0)})(u_0)$$

By the above construction, we can check that $\Gamma \in \mathcal{NO}_{N,\varphi,\psi}^{L,H,ReLU}$. Moreover, the depth $L(\Gamma)$ and the number $H(\Gamma)$ of neurons of the neural operator $\Gamma$ can be estimated as

$$L(\Gamma) \lesssim J \cdot L(F_{net}) \lesssim \log(\epsilon^{-1}) \cdot \log_2(\epsilon^{-1}) \lesssim (\log(\epsilon^{-1}))^2,$$

$$H(\Gamma) \lesssim J \cdot H(F_{net}) \lesssim \log(\epsilon^{-1}) \cdot \epsilon^{-1} \log_2(\epsilon^{-1}) \lesssim \epsilon^{-1} (\log(\epsilon^{-1}))^2.$$

Thus, the proof of Theorem 1 is complete. $\qquad \square$

# E    PROOF OF COROLLARY 1

By the triangle inequality, we estimate

$$\|u_{\max} - \hat{u}_{\max}\|_{L^r(0,\kappa^*T;L^s)} \leq \sum_{\kappa=1}^{\kappa^*} \|\Gamma^{+[\kappa]}(u_0) - \Gamma^{[\kappa]}(u_0)\|_{L^r((\kappa-1)T,\kappa T;L^s)}.$$

Therefore, as $\kappa^*$ is finite, it is enough to prove that

$$\begin{aligned}
&\|\Gamma^{+[\kappa]}(u_0) - \Gamma^{[\kappa]}(u_0)\|_{L^r((\kappa-1)T,\kappa T;L^s)} \\
&\leq C\epsilon + \|\Gamma^{+[\kappa-1]}(u_0)((\kappa-1)T) - \Gamma^{[\kappa-1]}(u_0)((\kappa-1)T)\|_{L^\infty}.
\end{aligned} \tag{33}$$

for some $C > 0$ and for $\kappa = 1, \ldots, \kappa^*$. For $\kappa = 1$, the estimate (33) is already obtained in Theorem 1. For $\kappa = 2, \ldots, \kappa^*$, we have

$$\begin{aligned}
&\|\Gamma^{+[\kappa]}(u_0) - \Gamma^{[\kappa]}(u_0)\|_{L^r((\kappa-1)T,\kappa T;L^s)} \\
&\leq \|\Gamma^+(\Gamma^{+[\kappa-1]}(u_0)) - \Gamma^+(\Gamma^{[\kappa-1]}(u_0))\|_{L^r((\kappa-1)T,\kappa T;L^s)} \\
&\quad + \|\Gamma^+(\Gamma^{[\kappa-1]}(u_0)) - \Gamma(\Gamma^{[\kappa-1]}(u_0))\|_{L^r((\kappa-1)T,\kappa T;L^s)}.
\end{aligned}$$

By Corollary 2 (or Proposition 1), the first term can be estimated as

$$\begin{aligned}
&\|\Gamma^+(\Gamma^{+[\kappa-1]}(u_0)) - \Gamma^+(\Gamma^{[\kappa-1]}(u_0))\|_{L^r((\kappa-1)T,\kappa T;L^s)} \\
&\leq C\|\Gamma^{+[\kappa-1]}(u_0)((\kappa-1)T) - \Gamma^{[\kappa-1]}(u_0)((\kappa-1)T)\|_{L^\infty}.
\end{aligned}$$

By Theorem 1, the second term in RHS can be estimated by $C\epsilon$ since $\Gamma^{[\kappa-1]}(u_0) \in B_{L^\infty}(R)$. Thus, the estimates (33) are proved for all $\kappa = 1, \ldots, \kappa^*$. $\qquad\square$

# F    PRESERVING THE POSITIVITY

We define the ReQU activation function by

$$ReQU(t) := \max\{0, t\}^2, \quad t \in \mathbb{R}.$$

**Corollary 5.** *Suppose that $\mathcal{L}$ and $F$ satisfy Assumptions 1, and 2, respectively, and the parameters $r, s$ satisfies the condition* (3). *Assume that $F \in C^1(\mathbb{R}; \mathbb{R})$ is a polynomial and satisfies that*

$$F(t) \lesseqgtr 0 \quad if\ t \lesseqgtr 0. \tag{34}$$

*Moreover, assume that there exists $N_0 \in \mathbb{N}$ such that the truncated expansion $G_N$ defined in Assumption 3 of the Green function $G$ satisfies*

$$G_N(t, x, y) \geq 0, \quad 0 < t < T,\ x, y \in D,$$

*for any $N \geq N_0$. Then, there exists $\Gamma \in \mathcal{NO}_{N,\varphi,\psi}^{L,H,ReQU}$ that the statement in Theorem 1 holds such that $\Gamma$ preserve the positivity , i.e.,*

$$\Gamma(u_0) \lesseqgtr 0 \quad if\ u_0 \in L^\infty(D)\ and\ u_0 \lesseqgtr 0.$$

*Proof.* We can show that there exist $\Gamma \in \mathcal{NO}_{N,\varphi,\psi}^{L,H,ReQU}$ satisfying the statement of Lemma 4 where $G_{N_k} \geq 0$ and $F_{net} : \mathbb{R} \to \mathbb{R}$ is ReQU neural network and exactly represents the polynomial $F$ (see e.g., Li & Yu (2019, Theorem 2.2)). Note that $\Gamma(u_0) = \Phi_{N_k,net}^{[J]}[0]$ where $\Phi_{N_k,net}$ is a map defined in (24) and depends on $u_0$. We will show that by induction

$$\Phi_{N_k,net}^{[J]}[0] \lesseqgtr 0 \text{ if } u_0 \lesseqgtr 0.$$

Let $u_0 \lesseqgtr 0$. First, we see that $\Phi_{N_k,net}[0] = \Phi_{N_k,net}^{[1]}[0] \lesseqgtr 0$. Assume that $\Phi_{N_k,net}^{[j]}[0] \lesseqgtr 0$ $(1 \leq j \leq J - 1)$. Since $F = F_{net}$ satisfies (34), we see that

$$\begin{aligned}
\Phi_{N_k,net}^{[j+1]}[0] &= \Phi_{N_k,net}\left[\Phi_{N_k,net}^{[j]}[0]\right] \\
&= \underbrace{\int_D G_{N_k}(t, x, y)u_0(y)dy}_{\lesseqgtr 0} + \underbrace{\int_0^t \int_D G_{N_k}(t - \tau, x, y)F_{net}(\Phi_{N_k,net}^{[j]}[0](\tau, y))d\tau dy}_{\lesseqgtr 0} \lesseqgtr 0.
\end{aligned}$$

Thus, the required result is proved. $\qquad\square$

# G    ADDITIONAL REMARKS

In this paper, we focused on parabolic PDEs with power-type (or somewhat general) nonlinear terms. Below, we provide additional remarks on Assumptions 1 and 2, and some remarks on applicability and limitation.

**Regarding Assumption 1.** Assumption 1 for $\mathcal{L}$ is a condition satisfied by the solution operators of many linear parabolic PDEs. In particular, it generalizes the rate of time $t$ using the parameter $\nu$, and it is known that many important operators $\mathcal{L}$ satisfy Assumption 1, as stated in Appendix A. On the other hand, There exist examples such that Assumption 1 does not hold. For instance, Schrödinger operators with inverse square potentials, i.e., $\mathcal{L} = -\Delta + \frac{c}{|x|^2}$ (where $c$ is a real number greater than the negative of the best constant of Hardy's inequality), are known to satisfy the inequality in Assumption 1 when $\nu = d/2$ and $d \geq 3$, provided that $\frac{2d}{d+2} \leq q_1 \leq q_2 \leq \frac{2d}{d-2}$ (this range is optimal. See Ioku et al. (2016) and Ioku & Ogawa (2019)). Thus, due to the constraints on the range of $q_1$ and $q_2$, certain operators may fail to satisfy Assumption 1. By extending the range of parameters in Assumption 1 to $q_{\min} \leq q_1 \leq q_2 \leq q_{\max}$, it is possible to generalize the assumption to include such examples. In this case, $q_{\min}$ and $q_{\max}$, in addition to $\nu$, would influence the conditions for well-posedness and approximation error estimates.

**Regarding Assumption 2.** There are various examples of nonlinear terms, such as $F(u) = |\nabla u|^p$ or $F(u) = u(e^{|u|^2} - 1)$, that do not satisfy Assumption 2. However, we believe that the arguments in this paper can be extended to such nonlinear terms as well. For example, the heat equation with $F(u) = |\nabla u|^p$, known as the viscous Hamilton-Jacobi equation, has its well-posedness established via a fixed-point argument (see Ben-Artzi et al. (2002)). Similarly, this is also true for $F(u) = u(e^{|u|^2} - 1)$ or more general exponential-type nonlinear terms (see Ibrahim et al. (2014)). The reason why our paper focused on Assumption 2 is because the results on well-posedness vary significantly depending on the nonlinear term and it is difficult to unify these results. Therefore, as a first step in this work, we focused on power-type nonlinear terms. As mentioned in Remark 1, smooth nonlinear terms can often be represented by their leading term in a power-type form through Taylor expansion, highlighting the importance of power-type nonlinear terms from this perspective.

**Regarding other equations.** We expect that similar arguments can be applied to a broader class of PDEs. For instance, initial(-boundary) value problems for various PDEs, such as the nonlinear Schrödinger equations and nonlinear wave equations (see Cazenave (2003); Cazenave & Haraux (1998)), nonlinear damped wave equations (see Ikeda et al. (2024)), the KPP-Fisher equation (see Huy & Tuan (2024)), the Allen-Cahn equation (see Israel (2013)), the Cahn-Hilirad equation (see Cholewa & Rodriguez-Bernal (2012)) and the Navier-Stokes equations (see Giga & Miyakawa (1985)), can be expressed as integral equations. By appropriately choosing the initial data space and the solution space, their well-posedness can be shown by using the fixed-point argument. On the other hand, for PDEs that involve nonlinearity in the principal part such as $p$-Laplacian equations $\partial_t u - \Delta_p u = F(u)$ with $\Delta_p := \text{div}(|\nabla u|^{p-2} \nabla u)$ and $p \neq 2$, it may not be possible to represent them in the form of integral equations. Such equations might not be effectively handled by our approach.

