# OpenReview forum: "Quantitative Approximation for Neural Operators in Nonlinear Parabolic Equations"
_ICLR.cc/2025/Conference — ICLR 2025 Poster_

### Official Review · Reviewer_pUKd · 2024-10-26

**Soundness:** 3
**Presentation:** 3
**Contribution:** 3
**Rating:** 5
**Confidence:** 3

**Summary:**

This paper studies the approximation property of the neural operator for representing the solution operator for nonlinear parabolic equations. The key idea is that, the solution of the equation, written as the Duhamel’s integral, can be obtained from Picard iteration, with exponential convergence. The authors use a deep neural operator to approximate the Picard iteration process, where each layer approximates one step of Picard iteration. Overall, the paper is well written with clear explanation for all the ideas.

**Strengths:**

Same as summary.

**Weaknesses:**

Need for clarification on ReLU approximation. Detail in Questions.

**Questions:**

My primary concern is with the ReLU network approximation, specifically in the correspondence between the neural operator structure defined on page 6 and Lemma 2. In Eq. (24), you define the mapping, but it remains unclear how this mapping aligns with the neural operator structure described earlier. Please explain how the neural operator structure in page 6 result in eq (24). A clearer explanation or justification is needed to ensure consistency between these sections.

Additionally, I noticed that the notation $\Phi$ represents one step (either exact or approximated) of the iteration. However, on page 6, within each iteration, the activation is only used once, which is a shallow neural network with a single hidden layer. The reference to Guhring et al. (2020, Theorem 4.1) pertains to deep neural networks, which might not directly apply to this context. I suggest finding a more appropriate reference for the approximation result, specifically one that addresses shallow networks.
Another minor question is that, I think your remark “the curse of dimensionality that occurs in conventional neural network approximations does not appear here” in page 8 does not make sense. As addressed by yourself, the curse of dimensionality is hidden in Assumption 3, the representation of the Green’s function of the linear equation.

Another minor question is that, I think your remark “the curse of dimensionality that occurs in conventional neural network approximations does not appear here” in page 8 does not make sense. As addressed by yourself, the curse of dimensionality is hidden in Assumption 3, the representation of the Green’s function of the linear equation.

---

> ### Author Response · Authors · 2024-11-22
>
> We appreciate the detailed suggestions, criticisms and endorsement of the reviewer. We address all of these below:
>
> > My primary concern is with the ReLU network approximation ... to ensure consistency between these sections.
>
> > Additionally, I noticed that the notation represents one step ... appropriate reference for the approximation result, specifically one that addresses shallow networks.
>
> We appreciate your point.
> Indeed, the current version did not provide sufficient explanation of how the neural operator defined in Definition 1 represents Eq. (24), despite its importance.
> To clarify, one step $ \Phi_{N, \text{net}} $ in Eq. (24) corresponds not to a single-layer of the neural operators, but to a multi-layer. Specifically, when the neural network $ F_{\text{net}} : \mathbb{R} \to \mathbb{R} $ has $ L $ layers, $ \Phi_{N, \text{net}} $ can represent a neural operator with $ L+1 $ layers. In this construction, the local operators $ W^{(\ell)} $ and biases $ b^{(\ell)} $ for $ \ell = 1, \dots, L $ correspond to those in the neural network $ F_{\text{net}} $, while the non-local operators $ K^{(\ell)} \equiv 0 $ for $ \ell = 1, \dots, L $. At the $ L+1 $-th layer, $ W^{(L+1)} \equiv 0 $ and $ b^{(L+1)} \equiv 0 $, and the non-local operator $ K^{(L+1)} $ corresponds to the integral operator used in Eq. (24).
> We have added these detailed explanations following Eq. (9) in the revised version.
>
>
> > Another minor question is that, I think your remark ... the Green’s function of the linear equation.
>
> We agree this sentence is not important. We removed it in the revised version.

---

### Official Review · Reviewer_HAsy · 2024-10-28

**Soundness:** 3
**Presentation:** 3
**Contribution:** 3
**Rating:** 6
**Confidence:** 4

**Summary:**

The paper defines a neural operator based on Picard iteration and provides a quantitative approximation theorem for learning the solution operator of a general class of non-linear parabolic PDEs. The paper starts with a local well-posedness result for the considered PDEs, then introduces the considered neural operators, states the quantitative approximation theorem, provides a sketch of the proof and discusses extensions to longer time-horizons. The supplementary material contains proofs and shows how the result can be applied to Fourier Neural Operators (FNOs) and Haar Wavelet Neural Operators (WNOs).

**Strengths:**

Overall, the paper is well-written and structured. The results hold in quite general settings: Assumption 1 on the differential operator $\mathcal{L}$ of the PDE is rather general and includes, e.g., Laplace operators with different types of boundary conditions. Assumption 2 on non-linearity seems more restrictive and mainly limited to power-type non-linearities, but the obtained results are useful even in the case $F=0$, which is also covered by the setting. Assumption 3 is crucial to make the approach work, based on an expansion of the underlying Green's function. In the supplementary material the paper specializes the setting to certain types of Fourier neural operators and Haar wavelet neural operators, demonstrating the usefulness of the result.

**Weaknesses:**

The setting is formulated very abstractly, making reading potentially difficult for readers less familiar with general PDE theory. The paper does not discuss how the constructed neural operators could be implemented. In relation to existing neural operators (FNOs, WNOs) it remains unclear in what generality they are covered. Relation to existing literature is not always clear. I make these points more specific in the questions posed below.

Generally, I think this could be a valuable contribution, enhancing our understanding of mathematical foundations of neural operators. However, currently there seems to be a mistake in the proof of Theorem 1, affecting the construction of the neural operators and potentially making implementation infeasible. Consequently, at this point I need to recommend rejection, but I would be willing to increase my rating if this issue can be addressed convincingly.

### Problem in the proof of Lemma 5

The main issue I currently see is that there appears to be a problem in the proof / formulation of the neural operators. The issue concerns the use of the time horizon $T$ instead of time variable $t$. In the definition of $\Phi_N$ in lines 264-269, in the first line the time integral is from $0$ to $t$ (the variable of the function to be approximated). In the next line, the integral with respect to $\tau$ is replaced by the inner product $\langle \psi_m, F(u) \rangle$. However, this means you are taking the integral up to $T$, not just up to $t$.
The same problem can be found in the proof of Lemma 5 in the Appendix (lines 1245-1248), which does not seem to be correct in its current form. It seems that, to address this, the neural operator architecture would need to be modified, by making also the inner products $\langle \psi_m, u \rangle$ in the definition of the operator $K^{(\ell)}_N$ depend on the time variable $t$.
However, this would make a crucial difference from the perspective of computational effort required to evaluate the neural operator. In the current notation, to evaluate  $K^{(\ell)}_N$  only one inner product needs to be evaluated (the same for all times $t$). It seems that to fix the issue mentioned above, for each time $t$ a different inner product would need to be evaluated, which may be computationally infeasible (as at any given hidden layer, you are computing inner products with output functions from previous hidden layers). Here a thorough discussion is required.
Even without this issue, a more thorough discussion on implementation would be needed: The approach crucially relies on evaluating inner products, by definition of  $K^{(\ell)}_N[u]$ . To compute such inner products (between the hidden layer functions and the basis functions $\psi_m$) in practice, additional discretization is required at each layer; for each function evaluation.

**Questions:**

### Additional points requiring clarification
1.	The paper mentions that Theorem 1 avoids the curse of dimensionality. However, this is not entirely correct: it seems that in the statement of the theorem, the involved constant $C$ may still grow exponentially in $d$?
2.	Theorem 1 requires $N$ to be chosen in such a way that $C_G(N) \leq \varepsilon$. Is it possible to quantitatively relate $N$ and $\varepsilon$, e.g., in the FNO case?
3.	P.3: You state that under suitable conditions on $\mathcal{L}$ and $F$, the problems (P) and (P’) are equivalent, if the function $u$ is sufficiently smooth. Could you give some precise references here?
4.	Appendix A does not fully clarify why the considered operators satisfy Assumption 1. For example, A.1 states a bound on the Green function, but does not make explicit why this bound implies Assumption 1. For readers less familiar with PDE theory it would be useful to make explicit (at least for one of the sections A.1, A.2, …) why this bound implies that Assumption 1 is satisfied.
5.	Similar results to Proposition 1 are well-known in the literature. Can you relate Proposition 1 to the literature more precisely, outlining differences to existing results in the literature?
6.	Fourier neural operators: why do you only consider the case $d=1$ here?
7.	Could you clarify if WNOs with Haar wavelets, as defined in Appendix C.2, have been used in previous papers?
8.	What is the role of $\tilde{D}$ in Appendix C.1? Can’t we just set $\tilde{D}=D$?


### Additional comments:

-P.4, l182: it should be operator, not operato,r

---

> ### Author Response · Authors · 2024-11-22
>
> We appreciate the detailed suggestions, criticisms and endorsement of the reviewer. We address all of these below:
>
> > Problem in the proof of Lemma 5.
> The main issue I currently  ... is required at each layer; for each function evaluation.
>
> We first empathize that our analysis does not have errors in the proof. Since our explanation was insufficient, we answer your question as follows:
>
> Suppose that we have the integral
>
> $$
> \int_{0}^{t}\int_{D} G(t-\tau, x, y) f(\tau, y) d\tau d y.
> $$
>
> By considering the extension $\tilde{G} : [0,T] \times [0,T] \times D \times D \to \mathbb{R}$ of $G$ by
>
> $$
> \tilde{G} (t,\tau,x,y) = G(t-\tau,x,y) \quad \text{if } 0 \le \tau < t < T, \ x,y \in D
> $$
> $$
>  \tilde{G} (t,\tau,x,y) =  0  \quad \text{otherwise},
> $$
>
> and expanding $\tilde{G}$ (regarding a function on $[0,T] \times [0,T] \times D  \times D$) using basis function $\varphi: [0,T] \times D \to \mathbb{R}$ and $\psi:[0,T] \times D \to \mathbb{R}$, we can see that
>
> $$
> \int_{0}^{t} \int_{D} G(\tau - t, x, y) f(\tau, y) d \tau dy
> $$
> $$
> = \int_0^{T}\int_D \tilde{G}(t, \tau, x, y) f(\tau, y) d\tau d y
> = \sum_{m,n\in\Lambda}
> c_{n,m} \left( \int_0^{T}\int_D \psi_{m}(\tau, y) f(\tau, y) d\tau d y \right)
> \varphi_{n}(t, x),
> $$
>
> where coefficients $c_{n,m}$ are given by
> $$
> c_{n,m} = \int_0^{T} \int_0^{T} \int_D \int_D  \tilde{G}(t, \tau, x, y) dt d\tau dx dy.
> $$
> Note that coefficients $c_{n,m}$ should be independent of input functions, especially time $t$, because coefficients $c_{n,m}$ correspond to learnable parameters in non-local operators.
> This is why we adapted the integral from $0$ to $T$, instead of the integral from $0$ to $t$, in the formulation of our neural operators.
> We added some explanation in the beginning of Section 3 in the revised version.
>
> Note that, due to the form of integral equation (P'), our neural operators need to take the integral with respect to time, which may be computationally expensive as your point.
> However, the previous work of Section 7.3 of [Kovachki et al, 2023] has employed the Fourier transforms with respect to time for computing FNOs, whose techniques might be useful for computing our neural operators.
> We added these discussions in Section 5 of the revised version.
>
>
> > 1. The paper mentions that Theorem 1 avoids ... may still grow exponentially in $d$ ?
>
> Our primary focus is not on avoiding the curse of dimensionality with respect to the spatial dimension $d$.
> Instead, we aim to control model complexity by providing effective bounds for the depth $L$, the number $H$ of neurons, and the rank $N$ when precision $\epsilon$ goes to zero.
> We have achieved this goal, ensuring that our rates do not grow exponentially.
> However, addressing the curse of dimensionality with respect to $d$ would also be an interesting direction for future works.
> In fact, the constant $C$ in Theorem 1 depends on $|D|$ (see Eq. (23)).
> For example, if $D$ is a hypercube with side length
> $R>1$, then
> then $|D| \approx R^d$, which grows exponentially as $d \to \infty$.
> We have noted this observation as a limitation of our work below Theorem 1 in the revised version.
>
>
>
> > 2. Theorem 1 requires ... e.g., in the FNO case?
>
> The approximation rate of basis expansions is generally determined by the choice of basis $\varphi, \psi$, the properties of the target function (function space), and the norm used to measure the error. There is an extensive body of work on this topic, necessitating a literature review. Here, we briefly outline some well-known results.
> In the case of Fourier bases on $d$-dimensional torus $\mathbb T^d$,
> it can be shown by Parseval's identity that the approximation rate in $L^2$-norm is $O(N^{-\sigma + d/2})$ ($\sigma>d/2$) if
> the target function belongs to the $L^2$ type Sobolev space $H^\sigma (\mathbb T^d)$ (see [Dya1995] for $L^p$-framework).
> As another example,
> for many wavelet bases, the same approximation rate $O(N^{-s/d})$ is known based on Jackson-type estimates (see Theorem 4.3.2 in [Coh2003]). Furthermore, there are cases where the rates differ between linear approximation and nonlinear approximation (wavelet approximation); in particular, when the target function is discountinuous (see Remark 4.2.5 in [Coh2003]).
> We added these discussions after Theorem 1 in the revised version.
>
> References
>
> [Coh2003] Cohen, Albert. Numerical analysis of wavelet methods. Elsevier, 2003.
>
> [Dya1995] Dyachenko, M. I., The rate of u-convergence of multiple Fourier series, Acta Mathematica Hungarica {\bf 68}, no. 1-2 (1995), 55--70.

---

> > ### Author Response · Authors · 2024-11-22
> >
> > > 3. P.3: You state that under suitable conditions ... Could you give some precise references here?
> >
> > For example, in cases of parabolic PDEs with $\mathcal L=-\Delta$, it follows from the smoothing effect of $S_{\mathcal L}(t)$ that the solution to (P') is a classical solution to (P) (see the argument in the proof of Theorem 1 in [BC1996]).
> > Even in the case where $\mathcal L$ is a more general operator satisfying Assumption 1,
> > a similar argument can be also done by replacing the differentiation in $x$ with the operator $\mathcal L$
> > (together with use of the techniques in the proofs of Lemmas 3.5 and 3.10 in [IMT2021] for instance).
> > We added this comment after (P') in the revised version.
> >
> > References
> >
> > [BC1996] Brezis, H. and  Cazenave, T., {\it A nonlinear heat equation with singular initial data}, Journal
> > D’Analyse Mathematique {\bf 68}, no. 1 (1996), 277--304.
> >
> > [IMT2021] Iwabuchi, T., Matsuyama, T. and Taniguchi, K., {\it Bilinear estimates in Besov spaces
> > generated by the Dirichlet Laplacian},  Journal of Mathematical Analysis and Applications {\bf 494}, no. 2 (2021), 124640.
> >
> > > 4. Appendix A does not fully clarify why ... why this bound implies that Assumption 1 is satisfied.
> >
> > For example, Section 5.1 in [ITW2024] briefly describes the proof of the operator norm estimate based on the Gaussian upper bound. For the reader's convenience, we added this proof in Appendix A of
> > the revised version.
> >
> > References
> >
> > [ITW2024] Ikeda, M., Taniguchi, K. and Wakasugi, Y., {\it Global existence and asymptotic behavior
> > for semilinear damped wave equations on measure spaces}, Evolution Equations and Control
> > Theory {\bf 13}, no. 4 (2024), 1101--1125.
> >
> >
> > > 5. Similar results to Proposition 1 are ... outlining differences to existing results in the literature?
> >
> >
> > Proposition 1 is a well-known result, and we think it has little mathematical novelty in itself. In particular, there is an extensive amount of literature for the case where $\mathcal L = -\Delta$. However, since we could not find any references that explicitly state and prove Proposition 1, we have provided a detailed description on it in this paper.
> >
> >
> > > 6. Fourier neural operators: why do you only consider the case $d=1$ here?
> >
> > From the parameter conditions in Theorem 1, we can take $r=s=2$ only if $d=1$. For this reason, the discussion on FNO becomes simpler in the case of $d=1$ (in particular, Plancherel's theorem can be utilized).  While a similar argument may also be applicable for $d\ge 2$, the argument becomes more complicated because it is difficult that the Fourier coefficients belong to the specified function space. Therefore, this paper omits this discussion. On the other hand, since estimating wavelet coefficients is also easier in $L^q$-framework, we deal with WNOs for general dimensions.
> >
> >
> > >7. Could you clarify if WNOs with Haar wavelets, as defined in Appendix C.2, have been used in previous papers?
> >
> > The previous papers such as [Gupta et al 2021] and [Tripura et al 2023] have not employed Haar wavelet in the context of learning solution operators for PDEs.
> > Appendix C.2 just chosen the Haar wavelet as a concrete example of basis expansion of the Green function, and the discussions choosing other wavelets could be possible.
> > We commented this after Corollary 4 in the revised version.
> >
> > >8. What is the role of $\tilde{D}$ in Appendix C.1? Can’t we just set ?
> >
> > Since the Green function $G$ generally does not satisfy the periodic boundary condition, we extended the domain $D$ to $\tilde D$ and applied zero extension $\tilde G$ of $G$ to ensure that $\tilde G$ satisfies periodic boundary conditions. This allows us to apply the Fourier series expansion to $\tilde G$. If $\mathcal L$ is the Laplacian with the periodic boundary condition, $G$ also satisfies the periodic boundary condition, and such arguments are unnecessary.

---

> ### Comment · Reviewer_HAsy · 2024-11-25
>
> I would like to thank the authors for the clarifications, these have addressed most of my concerns. Although it is still not fully clear to what extent these results are really applicable (FNO example is for 1-dimensional data, WNO example with Haar wavelets which have not been used in practice), I think the results provide a valuable contribution. I have increased my score accordingly.

---

### Official Review · Reviewer_kN84 · 2024-10-30

**Soundness:** 4
**Presentation:** 4
**Contribution:** 4
**Rating:** 8
**Confidence:** 4

**Summary:**

The authors derive the approximation rate of solution operators for the class of nonlinear parabolic partial differential equations (PDEs), contributing to the quantitative approximation theorem for solution operators of nonlinear PDEs. They show that neural operators can efficiently approximate these solution operators without the exponential growth in model complexity, thus strengthening the theoretical foundation of neural operators. An innovative link between Picard iteration and neural operators is found and the authors also leverage solid PDE theory (using Duhamel's principle). This work provides a very good theoretical foundation for neural operators (and generalizes previous works on Neural operators such as Fourier Neural Operators, etc.).

**Strengths:**

This is a highly rigorous and technical paper. The details are presented clearly and I am able to understand the authors' motivation and thought process in developing the theory. I think this is a great theoretical contribution that has solid foundations in PDE theory to better understand Neural Operators (of which there are varieties like Fourier, Wavelet, etc.) and unifies all of them in this fairly decent class of PDEs. I hope to see extensions of this work (that the authors mentioned there would be) to more general classes of PDEs but this is a significant start. Using neural operators as solvers for PDEs has a wide array of applications, particularly in scientific computing and other "AI4Science" domains. What is interesting is the method and analysis they use to bypass, by appropriately selecting the basis functions within the neural operator, the exponential growth in model complexity. They also do not seem to have too many unreasonable assumptions (this I'm not highly confident about).  Error rates with respect to truncation N make sense. There is a lot of scope for future explorations with different architectures and PDEs.

**Weaknesses:**

1. The paper could provide a bit more intuition and background on Neural Operators to be a bit more self-contained. This could be presented in the Supplementary.
2. There are no numerical experiments (but for this paper I do not think this is a big weakness given that I believe the main contribution of theory is solid). Nevertheless, it would be nice to see in future (if it is even possible to implement all these different basis expansions, etc.)
3. I do not think it is easy to find a Green's function or design one, which I see as the main drawback of the paper (if there is to be a practical extension for Deep Learning). Perhaps it is not that hard to find one for the linear parabolic PDE. (addressed in the Questions below).
4. Minor: typo on line 182 "elliptic operato,r "
I cannot think of any more now, but it depends on my questions below as well as I may not have understood everything.

**Questions:**

1. I do not think it is easy to find a Green's function or design one, which I see as the main drawback of the paper. Perhaps it is not that hard for the linear parabolic PDE. (addressed in the Questions below). How can one go about finding these (for specific cases or in general) or is this mainly constructive? Or do you only need to characterize the regularity of the Green function and how can you do that, if that is the case?
2.  Could you clarify if the error being log ((1/eps)^{-1}) implies exponential dependency?
3 (a). How can I interpret K^{(l)}_N? W are weights, b are biases, but I am not sure what K (the rank) is (unless it is specific to the Neural Operator?)
(b) Can you clarify (or add in the paper to make it more self-contained (see point 1 in Weakness) what is the rank of a neural operator? I do not believe it is defined in your paper (I checked other sources to understand).
4. It seems the first 2 sentences of your Discussion are contradictory: you mention it is a limitation that your method only applies to parabolic PDEs, but the next sentence says your approach is not restricted to parabolic PDEs. Perhaps you could clarify that it is not truly a limitation then, and is just a first step?
5. For the long-time solution, how reasonable is it that u(T) again falls into  Ball(L_\infty, R) ? Does that limit the expressiveness of the solution of the PDE?

---

> ### Author Response · Authors · 2024-11-22
>
> We appreciate the detailed suggestions, criticisms and endorsement of the reviewer. We address all of these below:
>
> > Minor: typo on line 182 "elliptic operato,r " ... well as I may not have understood everything.
>
> We thanks for finding typo. We modified this in the revised version.
>
> > I do not think it is easy to find a Green's function ... how can you do that, if that is the case?]
>
> As you pointed out, the construction or accurate approximation of the Green function is key to realizing the idea proposed in this study. If a good approximation of the Green  function is obtained, nonlinear solutions can be constructed using arguments based on Picard's iteration.
>
> There are various approaches to approximating or constructing the Green function depending on the situation. For example, in cases such as $\mathcal L = -\Delta$ on $\mathbb R^d$ or $[0,1]^d$ with the zero Dirichlet or Neumann boundary condition, the
> explicit representations of the Green functions are known and may suffice. However, in many cases where such explicit representations are unknown, suitable approximations of the Green function are necessary. In this study, we adopted an approach based on basis function approximations. For Fourier bases and wavelet bases, approximation theories are well-established, and using these, the Green function can be approximated (under an appropriate topology) in our setting.
>
> Additionally, approaches that do not rely on basis expansions can also be considered. For example, approximating the Green function using neural networks (e.g., DeepONet) and combining such approximators with the ideas proposed in this study may allow for effective approximation of solution operators for nonlinear PDEs. As described above, various approaches are possible depending on the situation, and we consider these to be valuable directions for future work.
>
> > Could you clarify if the error being log $((1/eps)^{-1})$ implies exponential dependency?
>
> In Theorem 1, the necessary depth $L$ is estimated by $L \approx  (\log (\epsilon^{-1} ))^2$, which is not exponentially, but squared logarithmically increasing. While, the necessary number $H$ of neurons is estimated by $H \approx \epsilon^{-1}$, which is polynomial increasing. We added this comment after Theorem 1 in the revised version.
>
>
> > How can I interpret $K^{(l)}_N$? ... unless it is specific to the Neural Operator?)
>
> $K^{(l)}_N$ are non-local operators which learns the non-local behavior in PDEs.
> We can interpret FNOs [Li et al, 2020a] as the special case when $\varphi$ and $\psi$ in $K^{(l)}_N$ are chosen as Fourier basis.
> We added more explanations of non-local operators $K^{(l)}_N$ after Definition 1 of the revised version.
>
> > Can you clarify (or add in the paper to ... (I checked other sources to understand).
>
> The rank of neural operators corresponds to the number $N$ of the truncation of basis expansion in non-local operators $K^{(\ell)}_N$.
> We already mentioned the rank $N$ in the last line of Definition 1.
>
>
> > It seems the first 2 sentences of ... and is just a first step?
>
> We agree that the first 2 sentences have contradictory.
> In the revised version, we removed the first sentence "A limitation of our method is that it applies only to parabolic PDEs that can be solved using the Banach’s fixed point theory", which is not a limitation of our works.
> In addition, we added the sentence ``In this paper we focus on parabolic PDEs with power-type (or somewhat general) nonlinear terms as a first step of study of neural operators based on Picard's iteration." in Section 5 of the revised version.
>
> > For the long-time solution, how reasonable is it that ...  expressiveness of the solution of the PDE?
>
> This strongly depends on the initial conditions and the nonlinear term. For example, consider the PDEs $\partial_t u - \Delta u = |u|^{p-1} u$. When $1<p \le 1+2/d$ (which is known as the Fujita exponent), it is known that any positive solution blows up in finite time. As a result, $u(T)$ will necessarily exceed $B_{L^\infty}(R)$ at some time.  On the other hand, when $p> 1+2/d$, if $ \lVert u_0 \rVert_{L^\infty}$ is taken sufficiently small, then the solution $u$ can be extended globally in time. In this case, $u(T)$ always remain within $B_{L^\infty}(R)$. Moreover, there is an extensive body of research on the criteria for determining global solutions and blow-up solutions of PDEs.

---

> > ### Comment · Reviewer_kN84 · 2024-11-24
> >
> > Thank you for the clarifications. I will maintain my score as I believe this work can lead to many other interesting explorations and perhaps even experimental results or validation by simulations.

---

### Official Review · Reviewer_r9yU · 2024-11-01

**Soundness:** 3
**Presentation:** 4
**Contribution:** 3
**Rating:** 5
**Confidence:** 5

**Summary:**

This research focuses on a new method for approximating the solution operators of nonlinear parabolic PDEs using neural operators. The authors aim to bridge the gap between theoretical understanding of neural operators and their practical application as PDE solvers by developing a quantitative approximation theorem. This theorem demonstrates that neural operators, specifically designed with a structure inspired by Picard's iteration, can approximate solutions of parabolic PDEs efficiently without experiencing the exponential growth in model complexity often associated with general operator learning. The authors provide examples of how their method can be applied to Fourier neural operators (FNOs) and wavelet neural operators (WNOs), highlighting the potential for this approach to solve a wide range of nonlinear PDEs.

**Strengths:**

1. The paper presents a quantitative approximation theorem for approximating solution operators of nonlinear parabolic PDEs using neural operators.
2. The authors show that the depth and number of neurons in their neural operators do not grow exponentially with the desired accuracy, potentially mitigating the "curse of parametric complexity."
3. The proof is constructive, leveraging the connection between Picard's iteration and the forward propagation through the layers of the neural operator.
4. The proposed framework may be potentially generalizable to other nonlinear PDEs solvable by Picard's iteration, including the Navier-Stokes, nonlinear Schrodinger, and nonlinear wave equations.

**Weaknesses:**

1. The theoretical results are not supported by numerical experiments, making it difficult to assess the practical performance and efficiency of the proposed neural operator architecture.
2. The paper focuses on a specific class of parabolic PDEs amenable to analysis via Banach's fixed point theorem. The applicability to more general PDEs, particularly the Navier-Stokes equation mentioned in the abstract, is not demonstrated.
3. The paper does not adequately address the challenge of handling periodic boundary conditions, a crucial aspect for applying FNOs.
4. The paper does not discuss the role of the trace operator and how to ensure sufficient regularity for the boundary conditions when defining the neural operator $\Gamma$.
5. The assumptions on the operator L and the nonlinearity F are quite strong, potentially limiting the scope of applicability.

**Questions:**

1. Can the authors provide numerical experiments to validate the theoretical results and compare the performance of their neural operator with traditional PDE solvers or other neural operator architectures?
2. The dual of $L^\infty$ is more complex than that of $L^1$. The authors should consider revising or removing this aspect throughout the paper.
3. Could the authors address how the proposed method handles periodic boundary conditions, which are essential for applying FNOs? Additionally, could they discuss the role of the trace operator and how to ensure sufficient regularity for the boundary conditions when defining the neural operator $\Gamma$?
4. Can the authors elaborate on how the proposed framework can be extended to handle more general PDEs, specifically the Navier-Stokes equations mentioned in the abstract with their specific projection operator and boundary conditions?
5. In PDE theory, extending local solutions to global solutions is a challenging and significant issue. Although the paper acknowledges the challenge of extending local solutions to global ones, it does not provide concrete solutions or insights. Could the authors elaborate on this challenge and clarify the practical relevance of Corollary 1, especially regarding its error estimate for long-time solutions?
6. How does the choice of basis functions $(\phi$ and $\psi)$ in the neural operator affect the approximation error rates? Could the authors provide specific examples to illustrate these rates for different basis functions?
7. Regularity of the solution space is crucial in numerical approximation. However, this paper does not incorporate this aspect into an error analysis. Could the authors characterize the impact of function regularity on the error estimate?

---

> ### Author Response · Authors · 2024-11-22
>
> We appreciate the detailed suggestions, criticisms and endorsement of the reviewer. We address all of these below:
>
> > The assumptions on the operator L and ... the scope of applicability.
> > 4. Can the authors elaborate on how the ... operator and boundary conditions?
>
> In this paper, we focused on parabolic PDEs with power-type (or somewhat general) nonlinear terms
> as a first step of study of neural operators based on Picard's iteration. However, we believe that
> our approach is not restricted to parabolic PDEs and can be adapted to a broader class of nonlinear PDEs, because the Banach’s fixed point theory is widely applicable to many nonlinear PDEs.
> Below, we provide additional remarks on Assumptions 1 and 2, and some remarks on applicability and limitation, which were added in Appendix G of the revised version.
>
> **Regarding Assumption 1:**
> Assumption 1 for $\mathcal{L}$ is a condition satisfied by the solution operators of many linear parabolic PDEs. In particular, it generalizes the rate of time $t$ using the parameter $\nu$, and it is known that many important operators $\mathcal{L}$ satisfy Assumption 1, as stated in Appendix~A.
>
> On the other hand, there exist examples such that Assumption 1 does not hold. For instance, Schrödinger operators with inverse square potentials, i.e., $\mathcal L = -\Delta + \frac{c}{|x|^2}$ (where $c$ is a real number greater than the negative of the best constant of Hardy's inequality), are known to satisfy the inequality in Assumption 1 when $\nu = d/2$ and $d \geq 3$, provided that $\frac{2d}{d+2} \leq q_1 \leq q_2 \leq \frac{2d}{d-2}$ (this range is optimal. See [IMSS2016] and [IO2019]). Thus, due to the constraints on the range of $q_1$ and $q_2$, certain operators may fail to satisfy Assumption 1. By extending the range of parameters in Assumption 1 to $q_{\min} \leq q_1 \leq q_2 \leq q_{\max}$, it is possible to generalize the assumption to include such examples. In this case, $q_{\min}$ and $q_{\max}$, in addition to $\nu$, would influence the conditions for well-posedness and approximation error estimates.
>
> **Regarding Assumption 2:**
> There are various examples of nonlinear terms, such as $F(u) = |\nabla u|^p$ or $F(u) = u (e^{|u|^2} - 1)$, that do not satisfy Assumption 2. However, we believe that the arguments in this paper can be extended to such nonlinear terms as well. For example, the heat equation with $F(u) = |\nabla u|^p$, known as the viscous Hamilton-Jacobi equation, has its well-posedness established via a fixed-point argument (see [BMS2002]). Similarly, this is also true for $F(u) = u (e^{|u|^2} - 1)$ or more general exponential-type nonlinear terms (see [IJMS2014]).
>
> The reason why our paper focused on Assumption 2 is because the results on well-posedness  vary significantly depending on the nonlinear term and it is difficult to unify these results. Therefore, as a first step in this work, we focused on power-type nonlinear terms. As mentioned in Remark~1, smooth nonlinear terms can often be represented by their leading term in a power-type form through Taylor expansion, highlighting the importance of power-type nonlinear terms from this perspective.
>
> **Regarding other equations:**
> We expect that similar arguments can be applied to a broader class of PDEs. For instance, initial(-boundary) value problems for various PDEs, such as the nonlinear Schr\"odinger equations and nonlinear wave equations (see [Caz2003], [CH1998]) and the Navier-Stokes equations (see [GM1985]), can be expressed as integral equations. By appropriately choosing the initial data space and the solution space, their well-posedness can be shown by using the fixed-point argument.
>
> On the other hand, for PDEs that involve nonlinearity in the principal part such as $p$-Laplacian equations $\partial_t u -\Delta_p u=F(u)$ with
> $\Delta_p := \operatorname{div} (|\nabla u|^{p-2} \nabla u)$ and $p\not = 2$, it may not be possible to represent them in the form of integral equations. Such equations might not be effectively handled by our approach.

---

> > ### Author Response · Authors · 2024-11-22
> >
> > References
> >
> > [BMS2002] Ben-Artzi et al, The local theory for viscous Hamilton--Jacobi equations in Lebesgue spaces, Journal de math{\'e}matiques pures et appliqu{\'e}es {\bf 81}, no. 4 (2002), 343--378.
> >
> > [Caz2003] Cazenave, Thierry, Semilinear Schr\"odinger Equations, Courant Lect. Notes Math. {\bf 10}, Amer. Math. Soc., 2003.
> >
> > [CH1998] Cazenave et al,, An introduction to semilinear evolution equations, Oxford University Press, vol. 13, 1998.
> >
> > [GM1985] Giga et al, T. Solutions in $L^r$ of the Navier-Stokes initial value proble, Arch. Rational Mech. Anal. {\bf 89} (1985), 267--281.
> >
> > [IJMS2014] Ibrahim et al, Local well posedness of a 2D semilinear heat equation, Bulletin of the Belgian Mathematical Society-Simon Stevin {\bf 21}, no. 3 (2014), 535--551.
> >
> > [IMSS2016] Ioku et al, $L^p$–$L^q$ estimates for homogeneous operators, Communications in Contemporary Mathematics {\bf 18}, no. 3 (2016), 1550037.
> >
> > [IO2019] Ioku et al, Critical dissipative estimate for a heat semigroup with a quadratic singular potential and critical exponent for nonlinear heat equations, Journal of Differential Equations {\bf 266}, no. 4 (2019), 2274--2293.
> >
> > >1. Can the authors provide numerical experiments ... or other neural operator architectures ?
> >
> > Our primary focus is to establish a solid theoretical foundation for neural operators, with experimental studies of our theoretical findings deferred to future work.
> > However, we acknowledge the importance of experimental studies and believe that our work provides a conceptual 'recipe' as our proof is constructive.
> > Specifically, our key approach involves the Picard iteration, which enables the neural operators constructed in our proof to apply the same operators repeatedly (Remark 3).
> > This weight-tied architecture not only reduces memory consumption but also integrates domain-specific knowledge of solutions in PDEs.
> >
> > > 2. The dual of $L^\infty$ is more complex ... this aspect throughout the paper.
> >
> > In this paper,
> > the exponent $r'$ of the $L^{r'}$ space means simply the H\"older conjugate derived from the H\"older inequality used in the proofs, and the dual space of $L^\infty$ is not used in this paper.
> >
> > > 3. Could the authors address how the proposed ... for applying FNOs?
> >
> > It is possible to theoretically handle periodic boundary conditions in our setting, because
> > the Laplacian $\mathcal L$ on the $d$-dimensional torus is one example of Assumption 1. In fact, the integral kernel of the semigroup generated by this $\mathcal L$ is known to have the explicit representation via the Sturm-Liouville decomposition, and by this fact, it can be checked that the inequality in Assumption 1 holds for this semigroup.
> >
> > Furthermore, it should be noted that FNOs can also be applied to non-periodic boundary conditions. In fact, in Appendix C.1, we applied FNOs by using a zero extension of the non-periodic target function to satisfy periodic boundary conditions. Additionally, the paper [Li et al., 2020a] on FNOs says ``Traditional Fourier methods work only with periodic boundary conditions. However, the Fourier neural operator does not have this limitation. This is due to the
> > linear transform $W$ (the bias term) which keeps the track of non-periodic boundary. As an example, the Darcy Flow and the time domain of Navier-Stokes have non-periodic boundary conditions, and the Fourier neural operator still learns the solution operator with excellent accuracy." (see the comment on Non-periodic boundary condition on page 9, Sections 5.2 and 5.5 in [Li et al., 2020a]).
> >
> >
> > > Additionally, could they discuss the role of ... when defining the neural operator ?
> >
> > We are not sure the meaning of this question. Could you add more explains of this question if you are not satisfied by the above answers ?
> >
> > > 5. In PDE theory, extending local solutions to ... especially regarding its error estimate for long-time solutions?
> >
> > As you pointed out, extending local solutions to global solutions is indeed a significant challenge in general. For our parabolic PDEs, the existence time $T$ depends only on $\lVert u_0 \rVert_{L^{\infty}}$
> > with respect to the initial data in Propositions 1 and 2, and hence, the criterion
> > "$
> > \sup_{0<t<T_{\max}} \lVert u(t) \rVert_{L^{\infty}} < \infty \Rightarrow T_{\max} = \infty
> > $"
> > holds, where $T_{\max}$ is the maximal existence time of the solution $u$.
> > Therefore, as stated in Section 4.2, if the $L^\infty$-norm of the extended solution at its final time is below $R$, the solution can be further extended, and this verification is easy. The same applies to the output of the neural operator. For the extended solution and the output, it is sufficient to compute the accumulated $L^\infty$-norm error, which is also easy to evaluate.

---

> > > ### Author Response · Authors · 2024-11-22
> > >
> > > > 6. How does the choice of basis functions and ... these rates for different basis functions?
> > >
> > > The approximation rate of basis expansions is generally determined by the choice of basis $\varphi, \psi$, the properties of the target function (function space), and the norm used to measure the error. There is an extensive body of work on this topic, necessitating a literature review. Here, we briefly outline some well-known results.
> > > In the case of Fourier bases on $d$-dimensional torus $\mathbb T^d$,
> > > it can be shown by Parseval's identity that the approximation rate in $L^2$-norm is $O(N^{-\sigma + d/2})$ ($\sigma>d/2$) if
> > > the target function belongs to the $L^2$ type Sobolev space $H^\sigma (\mathbb T^d)$ (see [Dya1995] for $L^p$-framework).
> > > As another example,
> > > for many wavelet bases, the same approximation rate $O(N^{-s/d})$ is known based on Jackson-type estimates (see Theorem 4.3.2 in [Coh2003]). Furthermore, there are cases where the rates differ between linear approximation and nonlinear approximation (wavelet approximation); in particular, when the target function is discountinuous (see Remark 4.2.5 in [Coh2003]).
> > > We added these discussions after Theorem 1 in the revised version.
> > >
> > > References
> > >
> > > [Coh2003] Cohen, Albert. Numerical analysis of wavelet methods. Elsevier, 2003.
> > >
> > > [Dya1995] Dyachenko, M. I., The rate of u-convergence of multiple Fourier series, Acta Mathematica Hungarica {\bf 68}, no. 1-2 (1995), 55--70.
> > >
> > > > 7. Regularity of the solution space is crucial... the impact of function regularity on the error estimate?
> > >
> > >
> > > We agree that considering the regularity of the solution space is very important.
> > > The current error estimates are based on $L^p$ spaces, but we expect that these results can be extended to Sobolev spaces since the local well-posedness, the basis expansion of the Green's function, and the approximation of nonlinearity using neural networks can be discussed on Sobolev spaces.
> > > We have incorporated these discussions in Section 5 of the revised version.

---

> > > > ### Comment · Reviewer_r9yU · 2024-11-23
> > > >
> > > > Thanks again for addressing my concerns. However, the authors have not fully addressed the issues I raised. First, the authors should provide at least a toy example to support their theoretical results. Additionally, the issues surrounding the periodization Green function, extension to Sobolev spaces, and long-time estimates remain nontrivial and require further clarification. Regarding the extensibility of the assumptions, it appears that most of the examples provided are of a reaction-type nature. I believe that including a convective term would significantly impact the proof and should be carefully addressed. Furthermore, in the case of the Navier-Stokes equations, the authors need to be cautious, as the divergence-free condition needs to be rigorously incorporated into the function space. For these reasons, I will maintain my original score.

---

> > > > > ### Author Response · Authors · 2024-11-25
> > > > >
> > > > > Thank you for your feedback.
> > > > > The reviewer's points are very interesting and we have discussed the potential extensions of our work to other equations, boundary conditions, and nonlinearities in the revised version, while we believe that the details requested by the reviewer are beyond the scope of this study.
> > > > > Additionally, we have provided concrete examples, including detailed assumptions on differential operators and nonlinearities, as well as specific choices of basis functions.
> > > > > We again emphasize that our neural operators (Definition 1) encompass FNOs and WNOs through specific choices of the basis functions $\varphi$ and $\psi$, as the Fourier basis and wavelet basis, respectively. These choices serve as toy examples to support our theoretical results, as FNOs and WNOs are widely utilized neural operators.

---

> > > > > > ### Author Response · Authors · 2024-11-25
> > > > > >
> > > > > > We recognize that the points raised by the reviewer are significant challenges, and we plan to address them as part of our future work. Below, we provide additional comments on some of the reviewer's points.
> > > > > >
> > > > > > **Extension to Sobolev spaces:**
> > > > > >
> > > > > > We except our results can be generalized to the framework of Sobolev spaces $H^s_q$ (or Besov spaces $B^s_{q,r}$). For simplicity, we comment only the case where $\mathcal L=-\Delta$.
> > > > > > To achieve this extension,
> > > > > > we think three key points need to be addressed:
> > > > > >
> > > > > > - (1) LWP in Sobolev spaces.
> > > > > > - (2) Basis expansion in Sobolev spaces.
> > > > > > - (3) Universal approximation for NNs in Sobolev spaces (approximation of nonlinear terms by neural networks).
> > > > > >
> > > > > >
> > > > > > **For (1),** we need the following estimates:
> > > > > >
> > > > > > - (a) $L^{q_1}$-$L^{q_2}$ estimates including the differential operators:
> > > > > > $$
> > > > > >     \lVert |\nabla|^s S_{-\Delta} (t) \rVert_{L^{q_1}\to L^{q_2}} \le C t^{-\frac{d}{2} (\frac{1}{q_1} - \frac{1}{q_2}) - \frac{s}{2}},
> > > > > > $$
> > > > > >     where $|\nabla|^s$ is the differential operator of fractional order $s\ge 0$. See e.g. Theorem 1.1 in [Iwa2017]. This result is established for Besov spaces but holds similarly for Sobolev spaces.
> > > > > >
> > > > > > - (b)
> > > > > >     Boundedness of composition operators in Sobolev spaces:
> > > > > > $$
> > > > > >      \lVert | F(u) \rVert_{H^s_q} \le C  \lVert u \rVert_{H^s_q}
> > > > > > $$
> > > > > >     (see, e.g., Theorem 1.1 in [Rib1998]),
> > > > > >     or the fractional Leibniz rule
> > > > > > $$
> > > > > >      \lVert |\nabla |^s (fg) \rVert_{L^{q}}
> > > > > >     \le C ( \lVert |\nabla|^s f \rVert_{L^{q_1}} \lVert g \rVert_{L^{q_2}} +  \lVert f \rVert_{L^{q_3}}  \lVert|\nabla |^s g \rVert_{L^{q_4}}),
> > > > > > $$
> > > > > >     where $1/q = 1/q_1 + 1/q_2 = 1/q_3 + 1/q_4$ (for natural numbers $p$, the power of nonlinearity, see [IMT2021] and references therein). By applying these results and using the standard fixed-point argument, the LWP in Sobolev spaces can be shown.
> > > > > >
> > > > > > **For (2),** if the target function is sufficiently smooth and the basis functions are either Fourier bases or smooth wavelets, the result can be similarly shown.
> > > > > >
> > > > > > **For (3),** many research results on universal approximation in Sobolev and Besov spaces can be utilized (see, e.g., Theorem 4.1 in [GKP2020]).
> > > > > >
> > > > > > Of course, we recognize that this problem is nontrivial, and the rough discussion provided here is insufficient. More rigorous analysis and a thorough literature review are required.
> > > > > >
> > > > > > **Long-time estimates:**
> > > > > > Regarding the approximation of long-time solutions or global-in-time solutions, a more equation-specific discussion would be required. For instance, as mentioned in our previous response, in the case of our nonlinear parabolic equation (with power-type nonlinear terms), the maximal existence time of the solution essentially depends on the size of the initial data. However, for example, in the cases of the Burgers equation and the viscous Hamilton-Jacobi equation, it is known that a solution exists globally in time for arbitrary initial data under appropriate settings.
> > > > > > In fact, the Burgers equation can be reduced to a linear problem through the Cole-Hopf transformation, and for the viscous Hamilton-Jacobi equation, the large data global existence was proved in [BMS2002]. For such equations that are globally stable in time, it might be possible to establish a more rigorous global-in-time approximation theory.
> > > > > >
> > > > > > **The case of the Navier-Stokes equations:**
> > > > > > For simplicity, we consider only the case of
> > > > > > the incompressible Navier-Stokes equation on $\mathbb R^d$. It is known that, through the Helmholtz-Weyl decomposition, the Cauchy problem of this equation can be reduced into
> > > > > > $$
> > > > > >     \partial_t u + Au + P(u\cdot \nabla u)=0,\quad t>0,
> > > > > > $$
> > > > > > $$
> > > > > >     u(0)=u_0,
> > > > > > $$
> > > > > > where $A=-P\Delta$ is the Stokes operator and $P$ is defined by the projection $P := (\delta_{jk} + R_j R_k )$ with
> > > > > > the Kronecker delta $\delta_{jk}$ and
> > > > > > the Riesz operator $R_j = (\partial/\partial x_j)(-\Delta)^{-1/2}$.
> > > > > > Then the Stokes semigroup $( S_A(t) )$ can be defined and it satisfies the estimates
> > > > > > $$
> > > > > > \lVert \nabla^l S_A(t) u_0 \rVert_{L^{q_2}}\le C t^{-\frac{d}{2} (\frac{1}{q_1}-\frac{1}{q_2}) -\frac{l}{2}} \lVert u_0 \rVert_{L^{q_1}},\quad t>0,\quad l=0,1
> > > > > > $$
> > > > > > for any $u_0 \in L^r$ with $\operatorname{div} u =0$ (note that this is a rough statement). Moreover, by the Duhamel principle, this problem can be reduced into the integral equation
> > > > > > $$
> > > > > > u(t) = S_A(t) u_0 + \int_0^t S_A(t-\tau) P( u \cdot \nabla u)(\tau)\, d\tau.
> > > > > > $$
> > > > > > Then the fixed point argument allows us to obtain the LWP for this integral equation (see e.g. [GM1985]). Therefore, there is a possibility that the idea presented in this paper is applicable to the incompressible Navier-Stokes equations, although the approximation of the Stokes semigroup and the projection remains a nontrivial challenge as the reviewer pointed out.  In particular, since these operators can be expressed as (usual or operator-valued) Fourier multiplier operators, it would be interesting to explore their connection to FNOs, which we consider as one of our future works.

---

### Meta-Review · Area_Chair_xQpA · 2024-12-18

**Metareview:**

This work studies the approximation properties of neural operators quantitatively for a broad class of nonlinear parabolic equations. The main results demonstrate that a class of neural operators, inspired by Picard's iteration, can approximate solution operators of parabolic PDEs efficiently without the exponential growth in model complexity. The authors provide examples of how their framework can be applied to Fourier neural operators and wavelet neural operators, making a concrete connection between theoretical findings and practice in operator learning.

Overall, the paper is well-organized and clearly written. The assumptions and their generality/limitations are carefully explained, and the separation of results into local time and long-time settings enhances the logical flow of the arguments. Overall, this work makes a strong theoretical contribution, rooted in PDE theory, while offering a fresh perspective on operator learning. As such, the AC believes this paper warrants acceptance as a rigorous theoretical study, even in the absence of numerical experiments.

**Additional Comments On Reviewer Discussion:**

The reviewers raised several points for clarification, including the conditions for long-time estimates, extensions to Sobolev spaces, details on the approximation of basis functions, and the irrelevance of the curse of dimensionality. During the discussion, the authors effectively addressed most of these concerns, either by providing detailed explanations or by outlining directions for future work. The paper was appropriately revised to incorporate this feedback and improve its overall clarity.

---

### Decision · Program_Chairs · 2025-01-22

Accept (Poster)